# Sample Complexity of Episodic Fixed-Horizon Reinforcement Learning

**Christoph Dann**
Machine Learning Department
Carnegie Mellon University
cdann@cdann.net

**Emma Brunskill**
Computer Science Department
Carnegie Mellon University
ebrun@cs.cmu.edu

## Abstract

Recently, there has been significant progress in understanding reinforcement learning in discounted infinite-horizon Markov decision processes (MDPs) by deriving tight sample complexity bounds. However, in many real-world applications, an interactive learning agent operates for a fixed or bounded period of time, for example tutoring students for exams or handling customer service requests. Such scenarios can often be better treated as episodic fixed-horizon MDPs, for which only looser bounds on the sample complexity exist. A natural notion of sample complexity in this setting is the number of episodes required to guarantee a certain performance with high probability (PAC guarantee). In this paper, we derive an upper PAC bound $\tilde{O}\big(\frac{|\mathcal{S}|^2|\mathcal{A}|H^2}{\epsilon^2}\ln\frac{1}{\delta}\big)$ and a lower PAC bound $\tilde{\Omega}\big(\frac{|\mathcal{S}||\mathcal{A}|H^2}{\epsilon^2}\ln\frac{1}{\delta+c}\big)$ that match up to log-terms and an additional linear dependency on the number of states $|\mathcal{S}|$. The lower bound is the first of its kind for this setting. Our upper bound leverages Bernstein's inequality to improve on previous bounds for episodic finite-horizon MDPs which have a time-horizon dependency of at least $H^3$.

## 1 Introduction and Motivation

Consider test preparation software that tutors students for a national advanced placement exam taken at the end of a year, or maximizing business revenue by the end of each quarter. Each individual task instance requires making a sequence of decisions for a fixed number of steps $H$ (e.g., tutoring one student to take an exam in spring 2015 or maximizing revenue for the end of the second quarter of 2014). Therefore, they can be viewed as a finite-horizon sequential decision making under uncertainty problem, in contrast to an infinite horizon setting in which the number of time steps is infinite. When the domain parameters (e.g. Markov decision process parameters) are not known in advance, and there is the opportunity to repeat the task many times (teaching a new student for each year's exam, maximizing revenue for each new quarter), this can be treated as episodic fixed-horizon reinforcement learning (RL). One important question is to understand how much experience is required to act well in this setting. We formalize this as the sample complexity of reinforcement learning [1], which is the number of time steps on which the algorithm may select an action whose value is not near-optimal. RL algorithms with a sample complexity that is a polynomial function of the domain parameters are referred to as Probably Approximately Correct (PAC) [2, 3, 4, 1]. Though there has been significant work on PAC RL algorithms for the infinite horizon setting, there has been relatively little work on the finite horizon scenario.

In this paper we present the first, to our knowledge, lower bound, and a new upper bound on the sample complexity of episodic finite horizon PAC reinforcement learning in discrete state-action spaces. Our bounds are tight up to log-factors in the time horizon $H$, the accuracy $\epsilon$, the number of actions $|\mathcal{A}|$ and up to an additive constant in the failure probability $\delta$. These bounds improve upon existing results by a factor of at least $H$. Our results also apply when the reward model is a function of the within-episode time step in addition to the state and action space. While we assume a stationary transition model, our results can be extended readily to time-dependent state-

transitions. Our proposed UCFH (Upper-confidence fixed-horizon RL) algorithm that achieves our upper PAC guarantee can be applied directly to wide range of fixed-horizon episodic MDPs with known rewards.[1] It does not require additional structure such as assuming access to a generative model [8] or that the state transitions are sparse or acyclic [6].

The limited prior research on upper bound PAC results for finite horizon MDPs has focused on different settings, such as partitioning a longer trajectory into fixed length segments [4, 1], or considering a sliding time window [9]. The tightest dependence on the horizon in terms of the number of episodes presented in these approaches is at least $H^3$ whereas our dependence is only $H^2$. More importantly, such alternative settings require the optimal policy to be stationary, whereas in general in finite horizon settings the optimal policy is nonstationary (e.g. is a function of both the state and the within episode time-step).[2] Fiechter [10, 11] and Reveliotis and Bountourelis [12] do tackle a closely related setting, but find a dependence that is at least $H^4$.

Our work builds on recent work [6, 8] on PAC infinite horizon discounted RL that offers much tighter upper and lower sample complexity bounds than was previously known. To use an infinite horizon algorithm in a finite horizon setting, a simple change is to augment the state space by the time step (ranging over $1, \ldots, H$), which enables the learned policy to be non-stationary in the original state space (or equivalently, stationary in the newly augmented space). Unfortunately, since these recent bounds are in general a quadratic function of the state space size, the proposed state space expansion would introduce at least an additional $H^2$ factor in the sample complexity term, yielding at least a $H^4$ dependence in the number of episodes for the sample complexity.

Somewhat surprisingly, we prove an upper bound on the sample complexity for the finite horizon case that only scales quadratically with the horizon. A key part of our proof is that the variance of the value function in the finite horizon setting satisfies a Bellman equation. We also leverage recent insights that state–action pairs can be estimated to different precisions depending on the frequency to which they are visited under a policy, extending these ideas to also handle when the policy followed is nonstationary. Our lower bound analysis is quite different than some prior infinite-horizon results, and involves a construction of parallel multi-armed bandits where it is required that the best arm in a certain portion of the bandits is identified with high probability to achieve near-optimality.

## 2   Problem Setting and Notation

We consider episodic fixed-horizon MDPs, which can be formalized as a tuple $M = (\mathcal{S}, \mathcal{A}, r, p, p_0, H)$. Both, the statespace $\mathcal{S}$ and the actionspace $\mathcal{A}$ are finite sets. The learning agent interacts with the MDP in episodes of $H$ time steps. At time $t = 1 \ldots H$, the agent observes a state $s_t$ and choses an action $a_t$ based on a policy $\pi$ that potentially depends on the within-episode time step, i.e., $a_t = \pi_t(s_t)$ for $t = 1, \ldots, H$. The next state is sampled from the stationary transition kernel $s_{t+1} \sim p(\cdot|s_t, a_t)$ and the initial state from $s_1 \sim p_0$. In addition the agent receives a reward drawn from a distribution[3] with mean $r_t(s_t)$ determined by the reward function. The reward function $r$ is possibly time-dependent and takes values in $[0, 1]$. The quality of a policy $\pi$ is evaluated by the *total expected reward* of an episode $R_M^\pi = \mathbb{E}\left[\sum_{t=1}^H r_t(s_t)\right]$. For simplicity,[1] we assume that the reward function $r$ is known to the agent but the transition kernel $p$ is unknown. The question we study is how many episodes does a learning agent follow a policy $\pi$ that is not $\epsilon$-optimal, i.e., $R_M^* - \epsilon > R_M^\pi$, with probability at least $1 - \delta$ for any chosen accuracy $\epsilon$ and failure probability $\delta$.

**Notation.**   In the following sections, we reason about the true MDP $M$, an empirical MDP $\hat{M}$ and an optimistic MDP $\tilde{M}$ which are identical except for their transition probabilities $p$, $\hat{p}$ and $\tilde{p}_t$. We will provide more details about these MDPs later. We introduce the notation explicitly only for $M$ but the quantities carry over to $\tilde{M}$ and $\hat{M}$ with additional tildes or hats by replacing $p$ with $\tilde{p}_t$ or $\hat{p}$.

The (linear) operator $P_i^\pi f(s) := \mathbb{E}[f(s_{i+1})|s_i = s] = \sum_{s' \in \mathcal{S}} p(s'|s, \pi_i(s))f(s')$ takes any function $f : \mathcal{S} \to \mathbb{R}$ and returns the expected value of $f$ with respect to the next time step.[4] For convenience, we define the multi-step version as $P_{i:j}^\pi f := P_i^\pi P_{i+1}^\pi \ldots P_j^\pi f$. The value function from time $i$ to time $j$ is defined as $V_{i:j}^\pi(s) := \mathbb{E}\left[\sum_{t=i}^j r_t(s_t)|s_i = s\right] = \sum_{t=i}^j P_{i:t-1}^\pi r_t = \left(P_i^\pi V_{i+1:j}^\pi\right)(s) + r_i(s)$ and $V_{i:j}^*$ is the optimal value-function. When the policy is clear, we omit the superscript $\pi$.

We denote by $\mathcal{S}(s, a) \subseteq \mathcal{S}$ the set of possible successor states of state $s$ and action $a$. The maximum number of them is denoted by $C = \max_{s,a \in \mathcal{S} \times \mathcal{A}} |\mathcal{S}(s, a)|$. In general, without making further assumptions, we have $C = |\mathcal{S}|$, though in many practical domains (robotics, user modeling) each state can only transition to a subset of the full set of states (e.g. a robot can't teleport across the building, but can only take local moves). The notation $\tilde{O}$ is similar to the usual $O$-notation but ignores log-terms. More precisely $f = \tilde{O}(g)$ if there are constants $c_1$, $c_2$ such that $f \leq c_1 g (\ln g)^{c_2}$ and analogously for $\tilde{\Omega}$. The natural logarithm is $\ln$ and $\log = \log_2$ is the base-2 logarithm.

## 3  Upper PAC-Bound

We now introduce a new model-based algorithm, UCFH, for RL in finite horizon episodic domains. We will later prove UCFH is PAC with an upper bound on its sample complexity that is smaller than prior approaches. Like many other PAC RL algorithms [3, 13, 14, 15], UCFH uses an optimism under uncertainty approach to balance exploration and exploitation. The algorithm generally works in phases comprised of optimistic planning, policy execution and model updating that take several episodes each. Phases are indexed by $k$. As the agent acts in the environment and observes $(s, a, r, s')$ tuples, UCFH maintains a confidence set over the possible transition parameters for each state-action pair that are consistent with the observed transitions. Defining such a confidence set that holds with high probability can be be achieved using concentration inequalities like the Hoeffding inequality. One innovation in our work is to use a particular new set of conditions to define the confidence set that enables us to obtain our tighter bounds. We will discuss the confidence sets further below. The collection of these confidence sets together form a class of MDPs $\mathcal{M}_k$ that are consistent with the observed data. We define $\hat{M}_k$ as the maximum likelihood estimate of the MDP given the previous observations.

Given $\mathcal{M}_k$, UCFH computes a policy $\pi^k$ by performing optimistic planning. Specifically, we use a finite horizon variant of extended value iteration (EVI) [5, 14]. EVI performs modified Bellman backups that are optimistic with respect to a given set of parameters. That is, given a confidence set of possible transition model parameters, it selects in each time step the model within that set that maximizes the expected sum of future rewards. Appendix A provides more details about fixed horizon EVI.

UCFH then executes $\pi^k$ until there is a state-action pair $(s, a)$ that has been visited often enough since its last update (defined precisely in the until-condition in UCFH). After updating the model statistics for this $(s, a)$-pair, a new policy $\pi^{k+1}$ is obtained by optimistic planning again. We refer to each such iteration of planning-execution-update as a *phase* with index $k$. If there is no ambiguity, we omit the phase indices $k$ to avoid cluttered notation.

UCFH is inspired by the infinite-horizon UCRL-$\gamma$ algorithm by Lattimore and Hutter [6] but has several important differences. First, the policy can only be updated at the end of an episode, so there is no need for explicit delay phases as in UCRL-$\gamma$. Second, the policies $\pi^k$ in UCFH are time-dependent. Finally, UCFH can directly deal with non-sparse transition probabilities, whereas UCRL-$\gamma$ only directly allows two possible successor states for each $(s, a)$-pair ($C = 2$).

**Confidence sets.**   The class of MDPs $\mathcal{M}_k$ consists of fixed-horizon MDPs $M'$ with the known true reward function $r$ and where the transition probability $p'_t(s'|s, a)$ from any $(s, a) \in \mathcal{S} \times \mathcal{A}$ to $s' \in \mathcal{S}(s, a)$ at any time $t$ is in the confidence set induced by $\hat{p}(s'|s, a)$ of the empirical MDP $\hat{M}$. Solely for the purpose of computationally more efficient optimistic planning, we allow time-dependent transitions (allows choosing different transition models in different time steps to maximize reward), but this does not affect the theoretical guarantees as the true stationary MDP is still in $\mathcal{M}_k$ with high

**Algorithm 1:** UCFH: **U**pper-**C**onfidence **F**ixed-**H**orizon episodic reinforcement learning algorithm

---

**Input**: desired accuracy $\epsilon \in (0,1]$, failure tolerance $\delta \in (0,1]$, fixed-horizon MDP $M$

**Result**: with probability at least $1-\delta$: $\epsilon$-optimal policy

$k := 1, \quad w_{\min} := \frac{\epsilon}{4H|\mathcal{S}|}, \quad \delta_1 := \frac{\delta}{2U_{\max}C}, \quad U_{\max} := |\mathcal{S} \times \mathcal{A}| \log_2 \frac{|\mathcal{S}|H}{w_{\min}};$

$m := 512(\log_2 \log_2 H)^2 \frac{CH^2}{\epsilon^2} \log^2 \left( \frac{8H^2|\mathcal{S}|^2}{\epsilon} \right) \ln \frac{6|\mathcal{S} \times \mathcal{A}|C \log_2^2(4|\mathcal{S}|^2 H^2/\epsilon)}{\delta};$

$n(s,a) = v(s,a) = n(s,a,s') := 0 \quad \forall, s \in \mathcal{S}, a \in \mathcal{A}, s' \in \mathcal{S}(s,a);$

**while do**

    /* Optimistic planning                                                  */

    $\hat{p}(s'|s,a) := n(s,a,s')/n(s,a)$, for all $(s,a)$ with $n(s,a) > 0$ and $s' \in \mathcal{S}(s,a)$;

    $\mathcal{M}_k := \big\{ \tilde{M} \in \mathcal{M}_{\text{nonst.}} : \forall (s,a) \in \mathcal{S} \times \mathcal{A}, t = 1 \ldots H, s' \in \mathcal{S}(s,a)$

                $\tilde{p}_t(s'|s,a) \in \texttt{ConfidenceSet}\,(\hat{p}(s'|s,a), n(s,a)) \big\};$

    $\tilde{M}_k, \pi^k := \texttt{FixedHorizonEVI}\,(\mathcal{M}_k);$

    /* Execute policy                                                    */

    **repeat**

        |   $\texttt{SampleEpisode}(\pi^k)$ ; // from $M$ using $\pi^k$

    **until** there is a $(s,a) \in \mathcal{S} \times \mathcal{A}$ with $v(s,a) \geq \max\{mw_{\min}, n(s,a)\}$ and $n(s,a) < |\mathcal{S}|mH$;

    /* Update model statistics for one $(s,a)$-pair with condition above      */

    $n(s,a) := n(s,a) + v(s,a);$

    $n(s,a,s') := n(s,a,s') + v(s,a,s') \quad \forall s' \in \mathcal{S}(s,a);$

    $v(s,a) := v(s,a,s') := 0 \quad \forall s' \in \mathcal{S}(s,a); k := k+1$

**Procedure** $\texttt{SampleEpisode}(\pi)$

    $s_0 \sim p_0;$

    **for** $t = 0$ **to** $H-1$ **do**

        $a_t := \pi_{t+1}(s_t)$ and $s_{t+1} \sim p(\cdot|s_t, a_t);$

        $v(s_t, a_t) := v(s_t, a_t) + 1$ and $v(s_t, a_t, s_{t+1}) := v(s_t, a_t, s_{t+1}) + 1;$

**Function** $\texttt{ConfidenceSet}\,(p, n)$

$$\mathcal{P} := \left\{ p' \in [0,1] \text{ :if } n > 1 : |p'(1-p') - p(1-p)| \leq \frac{2\ln(6/\delta_1)}{n-1}, \right. \tag{1}$$

$$\left. |p - p'| \leq \min\left( \sqrt{\frac{\ln(6/\delta_1)}{2n}}, \sqrt{\frac{2p(1-p)}{n}\ln(6/\delta_1)} + \frac{2}{3n}\ln\frac{6}{\delta_1} \right) \right\} \tag{2}$$

    **return** $\mathcal{P}$

---

probability. Unlike the confidence intervals used by Lattimore and Hutter [6], we not only include conditions based on Hoeffding's inequality[5] and Bernstein's inequality (Eq. 2), but also require that the variance $p(1-p)$ of the Bernoulli random variable associated with this transition is close to the empirical one (Eq. 1). This additional condition (Eq. 1) is key for making the algorithm directly applicable to generic MDPs (in which states can transition to any number of next states, e.g. $C > 2$) while only having a linear dependency on $C$ in the PAC bound.

### 3.1 PAC Analysis

For simplicity we assume that each episode starts in a fixed start state $s_0$. This assumption is not crucial and can easily be removed by additional notational effort.

**Theorem 1.** *For any $0 < \epsilon, \delta \leq 1$, the following holds. With probability at least $1 - \delta$, UCFH produces a sequence of policies $\pi^k$, that yield at most*

$$\tilde{O}\left( \frac{H^2 C|\mathcal{S} \times \mathcal{A}|}{\epsilon^2} \ln\frac{1}{\delta} \right)$$

*episodes with $R^* - R^{\pi^k} = V^*_{1:H}(s_0) - V^{\pi^k}_{1:H}(s_0) > \epsilon$. The maximum number of possible successor states is denoted by $1 < C \leq |\mathcal{S}|$.*

**Similarities to other analyses.** The proof of Theorem 1 is quite long and involved, but builds on similar techniques for sample-complexity bounds in reinforcement learning (see e.g. Brafman and Tennenholtz [3], Strehl and Littman [16]). The general proof strategy is closest to the one of UCRL-$\gamma$ [6] and the obtained bounds are similar if we replace the time horizon $H$ with the equivalent in the discounted case $1/(1-\gamma)$. However, there are important differences that we highlight now briefly.

- A central quantity in the analysis by Lattimore and Hutter [6] is the local variance of the value function. The exact definition for the fixed-horizon case will be given below. The key insight for the almost tight bounds of Lattimore and Hutter [6] and Azar et al. [8] is to leverage the fact that these local variances satisfy a Bellman equation [17] and so the discounted sum of local variances can be bounded by $O((1-\gamma)^{-2})$ instead of $O((1-\gamma)^{-3})$. We prove in Lemma 4 that local value function variances $\sigma_{i:j}^2$ also satisfy a Bellman equation for fixed-horizon MDPs even if transition probabilities and rewards are time-dependent. This allows us to bound the total sum of local variances by $O(H^2)$ and obtain similarly strong results in this setting.

- Lattimore and Hutter [6] assumed there are only two possible successor states (i.e., $C = 2$) which allows them to easily relate the local variances $\sigma_{i:j}^2$ to the difference of the expected value of successor states in the true and optimistic MDP $(P_i - \tilde{P}_i)\tilde{V}_{i+1:j}$. For $C > 2$, the relation is less clear, but we address this by proving a bound with tight dependencies on $C$ (Lemma C.6).

- To avoid super-linear dependency on $C$ in the final PAC bound, we add the additional condition in Equation (1) to the confidence set. We show that this allows us to upper-bound the total reward difference $R^* - R^{\pi^k}$ of policy $\pi^k$ with terms that either depend on $\sigma_{i:j}^2$ or decrease linearly in the number of samples. This gives the desired linear dependency on $C$ in the final bound. We therefore avoid assuming $C = 2$ which makes UCFH directly applicable to generic MDPs with $C > 2$ without the impractical transformation argument used by Lattimore and Hutter [6].

We will now introduce the notion of *knownness* and *importance* of state-action pairs that is essential for the analysis of UCFH and subsequently present several lemmas necessary for the proof of Theorem 1. We only sketch proofs here but detailed proofs for all results are available in the appendix.

**Fine-grained categorization of $(s, a)$-pairs.** Many PAC RL sample complexity proofs [3, 4, 13, 14] only have a binary notion of "knownness", distinguishing between known (transition probability estimated sufficiently accurately) and unknown $(s, a)$-pairs. However, as recently shown by Lattimore and Hutter [6] for the infinite horizon setting, it is possible to obtain much tighter sample complexity results by using a more fine grained categorization. In particular, a key idea is that in order to obtain accurate estimates of the value function of a policy from a starting state, it is sufficient to have only a loose estimate of the parameters of $(s, a)$-pairs that are unlikely to be visited under this policy.

Let the *weight* of a $(s, a)$-pair given policy $\pi^k$ be its expected frequency in an episode

$$w_k(s, a) := \sum_{t=1}^{H} \mathbb{P}(s_t = s, \pi_t^k(s_t) = a) = \sum_{t=1}^{H} P_{1:t-1} \mathbb{I}\{s = \cdot, a = \pi_t^k(s)\}(s_0).$$

The *importance* $\iota_k$ of $(s, a)$ is its relative weight compared to $w_{\min} := \frac{\epsilon}{4H|\mathcal{S}|}$ on a log-scale

$$\iota_k(s, a) := \min\left\{z_i \, : \, z_i \geq \frac{w_k(s, a)}{w_{\min}}\right\} \quad \text{where } z_1 = 0 \text{ and } z_i = 2^{i-2} \; \forall i = 2, 3, \dots.$$

Note that $\iota_k(s, a) \in \{0, 1, 2, 4, 8, 16 \dots\}$ is an integer indicating the influence of the state-action pair on the value function of $\pi^k$. Similarly, we define the *knownness*

$$\kappa_k(s, a) := \max\left\{z_i \, : \, z_i \leq \frac{n_k(s, a)}{m w_k(s, a)}\right\} \in \{0, 1, 2, 4, \dots\}$$

which indicates how often $(s, a)$ has been observed relative to its importance. The constant $m$ is defined in Algorithm 1. We can now categorize $(s, a)$-pairs into subsets

$$X_{k, \kappa, \iota} := \{(s, a) \in X_k \, : \, \kappa_k(s, a) = \kappa, \iota_k(s, a) = \iota\} \quad \text{and} \quad \bar{X}_k = \mathcal{S} \times \mathcal{A} \setminus X_k$$

where $X_k = \{(s, a) \in \mathcal{S} \times \mathcal{A} \, : \, \iota_k(s, a) > 0\}$ is the *active set* and $\bar{X}_k$ the set of state-action pairs that are very unlikely under the current policy. Intuitively, the model of UCFH is accurate if only few

$(s, a)$ are in categories with low knownness – that is, important under the current policy but have not been observed often so far. Recall that over time observations are generated under many policies (as the policy is recomputed), so this condition does not always hold. We will therefore distinguish between phases $k$ where $|X_{k,\kappa,\iota}| \leq \kappa$ for all $\kappa$ and $\iota$ and phases where this condition is violated. The condition essentially allows for only a few $(s, a)$ in categories that are less known and more and more $(s, a)$ in categories that are more well known. In fact, we will show that the policy is $\epsilon$-optimal with high probability in phases that satisfy this condition.

We first show the validity of the confidence sets $\mathcal{M}_k$.

**Lemma 1** (Capturing the true MDP whp.)**.** *$M \in \mathcal{M}_k$ for all $k$ with probability at least $1 - \delta/2$.*

*Proof Sketch.* By combining Hoeffding's inequality, Bernstein's inequality and the concentration result on empirical variances by Maurer and Pontil [18] with the union bound, we get that $p(s'|s, a) \in \mathcal{P}$ with probability at least $1 - \delta_1$ for a single phase $k$, fixed $s, a \in \mathcal{S} \times \mathcal{A}$ and fixed $s' \in \mathcal{S}(s, a)$. We then show that the number of model updates is bounded by $U_{\max}$ and apply the union bound. □

The following lemma bounds the number of episodes in which $\forall \kappa, \iota : |X_{k,\kappa,\iota}| \leq \kappa$ is violated with high probability.

**Lemma 2.** *Let $E$ be the number of episodes $k$ for which there are $\kappa$ and $\iota$ with $|X_{k,\kappa,\iota}| > \kappa$, i.e. $E = \sum_{k=1}^{\infty} \mathbb{I}\{\exists(\kappa, \iota) : |X_{k,\kappa,\iota}| > \kappa\}$ and assume that $m \geq \frac{6H^2}{\epsilon} \ln \frac{2E_{\max}}{\delta}$. Then $\mathbb{P}(E \leq 6NE_{\max}) \geq 1 - \delta/2$ where $N = |\mathcal{S} \times \mathcal{A}| m$ and $E_{\max} = \log_2 \frac{H}{w_{\min}} \log_2 |\mathcal{S}|$.*

*Proof Sketch.* We first bound the total number of times a fixed pair $(s, a)$ can be observed while being in a particular category $X_{k,\kappa,\iota}$ in all phases $k$ for $1 \leq \kappa < |\mathcal{S}|$. We then show that for a particular $(\kappa, \iota)$, the number of episodes where $|X_{k,\kappa,\iota}| > \kappa$ is bounded with high probability, as the value of $\iota$ implies a minimum probability of observing each $(s, a)$ pair in $X_{k,\kappa,\iota}$ in an episode. Since the observations are not independent we use martingale concentration results to show the statement for a fixed $(\kappa, \iota)$. The desired result follows with the union bound over all relevant $\kappa$ and $\iota$. □

The next lemma states that in episodes where the condition $\forall \kappa, \iota : |X_{k,\kappa,\iota}| \leq \kappa$ is satisfied and the true MDP is in the confidence set, the expected optimistic policy value is close to the true value. This lemma is the technically most involved part of the proof.

**Lemma 3** (Bound mismatch in total reward)**.** *Assume $M \in \mathcal{M}_k$. If $|X_{k,\kappa,\iota}| \leq \kappa$ for all $(\kappa, \iota)$ and $0 < \epsilon \leq 1$ and $m \geq 512 \frac{CH^2}{\epsilon^2} (\log_2 \log_2 H)^2 \log_2^2 \left( \frac{8H^2 |\mathcal{S}|^2}{\epsilon} \right) \ln \frac{6}{\delta_1}$. Then $|\tilde{V}_{1:H}^{\pi^k}(s_0) - V_{1:H}^{\pi^k}(s_0)| \leq \epsilon$.*

*Proof Sketch.* Using basic algebraic transformations, we show that $|p - \tilde{p}| \leq \sqrt{\tilde{p}(1 - \tilde{p})} O \left( \sqrt{\frac{1}{n} \ln \frac{1}{\delta_1}} \right) + O \left( \frac{1}{n} \ln \frac{1}{\delta_1} \right)$ for each $\tilde{p}, p \in \mathcal{P}$ in the confidence set as defined in Eq. 2. Since we assume $M \in \mathcal{M}_k$, we know that $p(s'|s, a)$ and $\tilde{p}(s'|s, a)$ satisfy this bound with $n(s, a)$ for all $s, a$ and $s'$. We use that to bound the difference of the expected value function of the successor state in $M$ and $\tilde{M}$, proving that $|(P_i - \tilde{P}_i)\tilde{V}_{i+1:j}(s)| \leq O \left( \frac{CH}{n(s,\pi(s))} \ln \frac{1}{\delta_1} \right) + O \left( \sqrt{\frac{C}{n(s,\pi(s))} \ln \frac{1}{\delta_1}} \right) \tilde{\sigma}_{i:j}(s)$, where the local variance of the value function is defined as $\sigma_{i:j}^2(s, a) := \mathbb{E} \left[ (V_{i+1:j}^{\pi}(s_{i+1}) - P_i^{\pi} V_{i+1:j}^{\pi}(s_i))^2 | s_i = s, a_i = a \right]$ and $\sigma_{i:j}^2(s) := \sigma_{i:j}^2(s, \pi_i(s))$. This bound then is applied to $|\tilde{V}_{1:H}(s_0) - V_{1:H}(s_0)| \leq \sum_{t=0}^{H-1} P_{1:t} |(P_t - \tilde{P}_t)\tilde{V}_{t+1:H}(s)|$. The basic idea is to split the bound into a sum of two parts by partitioning of the $(s, a)$ space by knownness, e.g. that is $(s_t, a_t) \in \bar{X}_{\kappa,\iota}$ for all $\kappa$ and $\iota$ and $(s_t, a_t) \in \bar{X}$. Using the fact that $w(s_t, a_t)$ and $n(s_t, a_t)$ are tightly coupled for each $(\kappa, \iota)$, we can bound the expression eventually by $\epsilon$. The final key ingredient in the remainder of the proof is to bound $\sum_{t=1}^{H} P_{1:t-1} \sigma_{t:H}(s)^2$ by $O(H^2)$ instead of the trivial bound $O(H^3)$. To this end, we show the lemma below. □

**Lemma 4.** *The variance of the value function defined as $\mathcal{V}_{i:j}^{\pi}(s) := \mathbb{E} \left[ \left( \sum_{t=i}^{j} r_t(s_t) - V_{i:j}^{\pi}(s_i) \right)^2 | s_i = s \right]$ satisfies a Bellman equation $\mathcal{V}_{i:j} = P_i \mathcal{V}_{i+1:j} + \sigma_{i:j}^2$ which gives $\mathcal{V}_{i:j} = \sum_{t=i}^{j} P_{i:t-1} \sigma_{t:j}^2$. Since $0 \leq \mathcal{V}_{1:H} \leq H^2 r_{\max}^2$, it follows that $0 \leq \sum_{t=1}^{j} P_{i:t-1} \sigma_{t:j}^2(s) \leq H^2 r_{\max}^2$ for all $s \in \mathcal{S}$.*

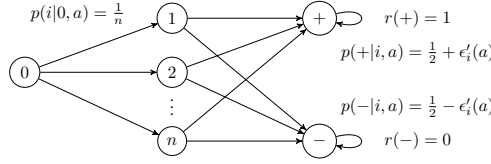

Figure 1: Class of a hard-to-learn finite horizon MDPs. The function $\epsilon'$ is defined as $\epsilon'(a_1) = \epsilon/2$, $\epsilon'(a_i^*) = \epsilon$ and otherwise $\epsilon'(a) = 0$ where $a_i^*$ is an unknown action per state $i$ and $\epsilon$ is a parameter.

*Proof Sketch.* The proof works by induction and uses fact that the value function satisfies the Bellman equation and the tower-property of conditional expectations. $\square$

**Proof Sketch for Theorem 1.** The proof of Theorem 1 consists of the following major parts:

1. The true MDP is in the set of MDPs $\mathcal{M}_k$ for all phases $k$ with probability at least $1 - \frac{\delta}{2}$ (Lemma 1).

2. The `FixedHorizonEVI` algorithm computes a value function whose optimistic value is higher than the optimal reward in the true MDP with probability at least $1 - \delta/2$ (Lemma A.1).

3. The number of episodes with $|X_{k,\kappa,\iota}| > \kappa$ for some $\kappa$ and $\iota$ are bounded with probability at least $1 - \delta/2$ by $\tilde{O}(|\mathcal{S} \times \mathcal{A}| \, m)$ if $m = \tilde{\Omega}\left(\frac{H^2}{\epsilon} \ln \frac{|\mathcal{S}|}{\delta}\right)$ (Lemma 2).

4. If $|X_{k,\kappa,\iota}| \leq \kappa$ for all $\kappa$, $\iota$, i.e., relevant state-action pairs are sufficiently known and $m = \tilde{\Omega}\left(\frac{CH^2}{\epsilon^2} \ln \frac{1}{\delta_1}\right)$, then the optimistic value computed is $\epsilon$-close to the true MDP value. Together with part 2, we get that with high probability, the policy $\pi^k$ is $\epsilon$-optimal in this case.

5. From parts 3 and 4, with probability $1 - \delta$, there are at most $\tilde{O}\left(\frac{C|\mathcal{S} \times \mathcal{A}|H^2}{\epsilon^2} \ln \frac{1}{\delta}\right)$ episodes that are not $\epsilon$-optimal.

## 4 Lower PAC Bound

**Theorem 2.** *There exist positive constants $c_1$, $c_2$, $\delta_0$, $\epsilon_0$ such that for every $\delta \in (0, \delta_0)$ and $\epsilon \in (0, \epsilon_0)$ and for every algorithm A that satisfies a PAC guarantee for $(\epsilon, \delta)$ and outputs a deterministic policy, there is a fixed-horizon episodic MDP $M_{hard}$ with*

$$\mathbb{E}[n_A] \geq \frac{c_1(H-2)^2(|\mathcal{A}|-1)(|\mathcal{S}|-3)}{\epsilon^2} \ln\left(\frac{c_2}{\delta + c_3}\right) = \Omega\left(\frac{|\mathcal{S} \times \mathcal{A}|H^2}{\epsilon^2} \ln\left(\frac{c_2}{\delta + c_3}\right)\right) \quad (3)$$

*where $n_A$ is the number of episodes until the algorithm's policy is $(\epsilon, \delta)$-accurate. The constants can be set to $\delta_0 = \frac{e^{-4}}{80} \approx \frac{1}{5000}$, $\epsilon_0 = \frac{H-2}{640e^4} \approx H/35000$, $c_2 = 4$ and $c_3 = e^{-4}/80$.*

The ranges of possible $\delta$ and $\epsilon$ are of similar order than in other state-of-the-art lower bounds for multi-armed bandits [19] and discounted MDPs [14, 6]. They are mostly determined by the bandit result by Mannor and Tsitsiklis [19] we build on. Increasing the parameter limits $\delta_0$ and $\epsilon_0$ for bandits would immediately result in larger ranges in our lower bound, but this was not the focus of our analysis.

*Proof Sketch.* The basic idea is to show that the class of MDPs shown in Figure 1 require at least a number of observed episodes of the order of Equation (3). From the start state 0, the agent ends up in states 1 to $n$ with equal probability, independent of the action. From each such state $i$, the agent transitions to either a good state $+$ with reward 1 or a bad state $-$ with reward 0 and stays there for the rest of the episode. Therefore, each state $i = 1, \ldots, n$ is essentially a multi-armed bandit with binary rewards of either 0 or $H - 2$. For each bandit, the probability of ending up in $+$ or $-$ is equal except for the first action $a_1$ with $p(s_{t+1} = +|s_t = i, a_t = a_1) = 1/2 + \epsilon/2$ and possibly an unknown optimal action $a_i^*$ (different for each state $i$) with $p(s_{t+1} = +|s_t = i, a_t = a_i^*) = 1/2 + \epsilon$.

In the episodic fixed-horizon setting we are considering, taking a suboptimal action in one of the bandits does not necessarily yield a suboptimal episode. We have to consider the average over all bandits instead. In an $\epsilon$-optimal episode, the agent therefore needs to follow a policy that would solve at least a certain portion of all $n$ multi-armed bandits with probability at least $1 - \delta$. We show that the best strategy for the agent to achieve this is to try to solve all bandits with equal probability. The number of samples required to do so then results in the lower bound in Equation (3). $\square$

Similar MDPs that essentially solve multiple of such multi-armed bandits have been used to prove lower sample-complexity bounds for discounted MDPs [14, 6]. However, the analysis in the infinite horizon case as well as for the sliding-window fixed-horizon optimality criterion considered by Kakade [4] is significantly simpler. For these criteria, every time step the agent follows a policy that is not $\epsilon$-optimal counts as a "mistake". Therefore, every time the agent does not pick the optimal arm in any of the multi-armed bandits counts as a mistake. This contrasts with our fixed-horizon setting where we must instead consider taking an average over all bandits.

## 5 Related Work on Fixed-Horizon Sample Complexity Bounds

We are not aware of any lower sample complexity bounds beyond multi-armed bandit results that directly apply to our setting. Our upper bound in Theorem 1 improves upon existing results by at least a factor of $H$. We briefly review those existing results in the following.

**Timestep bounds.** Kakade [4, Chapter 8] proves upper and lower PAC bounds for a similar setting where the agent interacts indefinitely with the environment but the interactions are divided in segments of equal length and the agent is evaluated by the expected sum of rewards until the end of each segment. The bound states that there are not more than $\tilde{O}\left(\frac{|\mathcal{S}|^2|\mathcal{A}|H^6}{\epsilon^3}\ln\frac{1}{\delta}\right)$[6] time steps in which the agents acts $\epsilon$-suboptimal. Strehl et al. [1] improves the state-dependency of these bounds for their delayed Q-learning algorithm to $\tilde{O}\left(\frac{|\mathcal{S}||\mathcal{A}|H^5}{\epsilon^4}\ln\frac{1}{\delta}\right)$. However, in episodic MDP it is more natural to consider performance on the entire episode since suboptimality near the end of the episode is no issue as long as the total reward on the entire episode is sufficiently high. Kolter and Ng [9] use an interesting sliding-window criterion, but prove bounds for a Bayesian setting instead of PAC. Timestep-based bounds can be applied to the episodic case by augmenting the original statespace with a time-index per episode to allow resets after $H$ steps. This adds $H$ dependencies for each $|\mathcal{S}|$ in the original bound which results in a horizon-dependency of at least $H^6$ of these existing bounds. Translating the regret bounds of UCRL2 in Corollary 3 by Jaksch et al. [20] yields a PAC-bound on the number of episodes of at least $\tilde{O}\left(\frac{|\mathcal{S}|^2|\mathcal{A}|H^3}{\epsilon^2}\ln\frac{1}{\delta}\right)$ even if one ignores the reset after $H$ time steps. Timestep-based lower PAC-bounds cannot be applied directly to the episodic reward criterion.

**Episode bounds.** Similar to us, Fiechter [10] uses the value of initial states as optimality-criterion, but defines the value w.r.t. the $\gamma$-discounted infinite horizon. His results of order $\tilde{O}\left(\frac{|\mathcal{S}|^2|\mathcal{A}|H^7}{\epsilon^2}\ln\frac{1}{\delta}\right)$ episodes of length $\tilde{O}(1/(1-\gamma)) \approx \tilde{O}(H)$ are therefore not directly applicable to our setting. Auer and Ortner [5] investigate the same setting as we and propose a UCB-type algorithm that has no-regret, which translates into a basic PAC bound of order $\tilde{O}\left(\frac{|\mathcal{S}|^{10}|\mathcal{A}|H^7}{\epsilon^3}\ln\frac{1}{\delta}\right)$ episodes. We improve on this bound substantially in terms of its dependency on $H$, $|\mathcal{S}|$ and $\epsilon$. Reveliotis and Bountourelis [12] also consider the episodic undiscounted fixed-horizon setting and present an efficient algorithm in cases where the transition graph is acyclic and the agent knows for each state a policy that visits this state with a known minimum probability $q$. These assumptions are quite limiting and rarely hold in practice and their bound of order $\tilde{O}\left(\frac{|\mathcal{S}||\mathcal{A}|H^4}{\epsilon^2 q}\ln\frac{1}{\delta}\right)$ explicitly depends on $1/q$.

## 6 Conclusion

We have shown upper and lower bounds on the sample complexity of episodic fixed-horizon RL that are tight up to log-factors in the time horizon $H$, the accuracy $\epsilon$, the number of actions $|\mathcal{A}|$ and up to an additive constant in the failure probability $\delta$. These bounds improve upon existing results by a factor of at least $H$. One might hope to reduce the dependency of the upper bound on $|\mathcal{S}|$ to be linear by an analysis similar to Mormax [7] for discounted MDPs which has sample complexity linear in $|\mathcal{S}|$ at the penalty of additional dependencies on $H$. Our proposed UCFH algorithm that achieves our PAC bound can be applied to directly to a wide range of fixed-horizon episodic MDPs with known rewards and does not require additional structure such as sparse or acyclic state transitions assumed in previous work. The empirical evaluation of UCFH is an interesting direction for future work.

**Acknowledgments:** We thank Tor Lattimore for the helpful suggestions and comments. This work was supported by an NSF CAREER award and the ONR Young Investigator Program.

## Footnotes

[1] Previous works [5] have shown that the complexity of learning state transitions usually dominates learning reward functions. We therefore follow existing sample complexity analyses [6, 7] and assume known rewards for simplicity. The algorithm and PAC bound can be extended readily to the case of unknown reward functions.

[2]The best action will generally depend on the state and the number of remaining time steps. In the tutoring example, even if the student has the same state of knowledge, the optimal tutor decision may be to space practice if there is many days till the test and provide intensive short-term practice if the test is tomorrow.

[3]It is straightforward to have the reward depend on the state, or state/action or state/action/next state.

[4]The definition also works for time-dependent transition probabilities.

[5]The first condition in the $\min$ in Equation (2) is actually not necessary for the theoretical results to hold. It can be removed and all $6/\delta_1$ can be replaced by $4/\delta_1$.

[6]For comparison we adapt existing bounds to our setting. While the original bound stated by Kakade [4] only has $H^3$, an additional $H^3$ comes in through $\epsilon^{-3}$ due to different normalization of rewards.

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
