[Supplementary Material · dann_brunskill_2015_full.pdf]

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

[7]While the considered random variables are strictly speaking not necessarily independent, they can be treated as such for the concentration inequalities applied here. See Appendix A of Strehl and Littman [16] for details.

[8]The empirical variance denoted by $V_n(\mathbf{X})$ by Maurer and Pontil [18] is $\tilde{p}(s'|s, a)(1 - \tilde{p}(s'|s, a))$ in our case and $\mathbb{E}V_n$ is the true variance which amounts to $p(s'|s, a)(1 - p(s'|s, a))$ for us.

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

# Appendices

# A Fixed-Horizon Extended Value Iteration

We want to find a policy $\pi^k$ and optimistic $\tilde{M}_k \in \mathcal{M}_k$ which have the highest total reward $R^{\pi^k}_{\tilde{M}_k} = \max_{\pi, M' \in \mathcal{M}_k} R^{\pi}_{M'}$. Note that $\pi^k$ is an optimal policy for $M_k$ but not necessarily for $M$. We can find such a policy by dynamic programming similar to extended value iteration [16, 5]. The optimal Q-function can be computed as $\tilde{Q}^*_{H:H}(s, a) = r_H(s)$ and for $i = H - 1, \dots, 2, 1$ as

$$\tilde{Q}^*_{i:H}(s, a) = r_i(s) + \max_{\tilde{p}_i \in \mathcal{P}_{s,a}} \left\{ \sum_{s' \in \mathcal{S}(s,a)} \tilde{p}_i \max_{b \in \mathcal{A}} \tilde{Q}^*_{i+1:H}(s', b) \right\}$$

The feasible set is defined as $\mathcal{P}_{s,a} := \{p \in [0,1]^{|\mathcal{S}(s,a)|} \mid \|p\|_1 = 1, \forall s' \in \mathcal{S}(s,a) : p(s') \in$ ConfidenceSet$(\hat{p}(s'|s,a), n(s,a))\}$. The optimal policy $\pi^k_t(s)$ at time $t$ is then simply the maximizer of the inner $\max$ operator and the transition probability $\tilde{p}_t(\cdot|s,a)$ is the maximizer of the outer maximum. The inner $\max$ can be solved efficiently by enumeration and the outer maximum similar to extended value iteration [16]. The basic idea is to put as much probability mass as possible to successor states with highest value. See the following algorithm for the implementation details.

---

**Function** FixedHorizonEVI($\mathcal{M}$)

$\tilde{Q}^*_{H:H}(s,a) = r_H(s) \quad \forall s, a \in \mathcal{S} \times \mathcal{A}$ ;  // $O(|\mathcal{S}||\mathcal{A}|)$
**for** $t = H - 1$ **to** $1$ **do**  // $O(H|\mathcal{S}|\log|\mathcal{S}| + H|\mathcal{S}||\mathcal{A}|C))$
   $\pi_{t+1}(s) := \arg\max_{a \in \mathcal{A}} \tilde{Q}^*_{t+1:H}(s,a) \quad \forall s \in \mathcal{S}$ ;  // $O(|\mathcal{S}||\mathcal{A}|)$
   sort states $s^{(1)}, \dots s^{(|\mathcal{S}|)}$ such that
      $\tilde{Q}^*_{t+1:H}(s^{(i)}, \pi_{t+1}(s^{(i)})) \geq \tilde{Q}^*_{t+1:H}(s^{(i+1)}, \pi_{t+1}(s^{(i+1)}))$ ;  // $O(|\mathcal{S}|\log|\mathcal{S}|)$
   **for** $s, a \in \mathcal{S} \times \mathcal{A}$ **do**  // $O(|\mathcal{S}||\mathcal{A}|C)$
      $\tilde{p}_t(s'|s,a) := \min$ ConfidenceSet$(\hat{p}(s'|s,a), n(s,a)) \quad \forall s' \in \mathcal{S}(s,a)$ ;  // $O(C)$
      $\Delta := 1 - \sum_{s' \in \mathcal{S}(s,a)} \tilde{p}_t(s'|s,a)$ ;  // $O(C)$
      $i := 1$ ;  // $O(1)$
      **while** $\Delta > 0$ **do**  // $O(C)$
         $s' := s^{(i)}$;
         $\Delta' := \min\{\Delta, \max$ ConfidenceSet$(\hat{p}(s'|s,a), n(s,a)) - \tilde{p}_t(s'|s,a)\}$;
         $\tilde{p}_t(s'|s,a) := \tilde{p}_t(s'|s,a) + \Delta$;
         $\Delta := \Delta - \Delta'; i := i + 1$;
      $\tilde{Q}^*_{t:H}(s,a) = \sum_{s' \in \mathcal{S}(s,a)} \tilde{p}_t(s'|s,a) \tilde{Q}^*_{t+1:H}(s', \pi_{t+1}(s'))$ ;  // $O(C)$

$\pi_1(s) := \arg\max_{a \in \mathcal{A}} \tilde{Q}^*_{1:H}(s,a) \quad \forall s \in \mathcal{S}$ ;  // $O(|\mathcal{S}||\mathcal{A}|$
**return** *MDP with transition probabilities $\tilde{p}_t$, optimal policy $\pi$*

---

Note that due to the nonlinear constraint in Equation (1), ConfidenceSet$(\hat{p}(s'|s,a), n(s,a))$ may be the union of two disjoint intervals instead of one interval. Still, $\min$- and $\max$-operations on the confidence sets can be computed readily in constant time. Therefore, the transition probabilities $\tilde{p}_t(\cdot|s,a)$ for a single time step $t$ and state-action pair $s, a$ can be computed in $O(|\mathcal{S}||\mathcal{A}|C)$ given sorted states. Sorting the states takes $O(|\mathcal{S}|\log|\mathcal{S}|)$ which results in $O(H|\mathcal{S}|\log|\mathcal{S}| + H|\mathcal{S}||\mathcal{A}|C)$ runtime complexity of FixedHorizonEVI (see comments in Function FixedHorizonEVI ). The Algorithm requires $O(H|\mathcal{S}||\mathcal{A}|C)$ additional space besides the storage requirements of the input MDP $\mathcal{M}$ as the transition probabilities $\tilde{p}_t$ are returned by the algorithm. If those are not required and only the optimal policy is of interest, the additional space can be reduced to $O(|\mathcal{S}||\mathcal{A}|)$.

**Lemma A.1** (Validity of optimistic planning). FixedHorizonEVI($\mathcal{M}_k$) *returns* $\tilde{M}, \pi^k = \arg\max_{M \in \mathcal{M}_k, \pi} R^{\pi}_M$.

*Proof Sketch.* This result can be proved straight-forwardly by showing that $\pi^k$ is optimal in the last time step $H$ with highest possible reward and then subsequently for all previous time steps inductively. It follows directly from the definition of the algorithm in Function FixedHorizonEVI that the returned MDP is in $\mathcal{M}_k$. □

# B    Runtime- and Space-Complexity of UCFH

Sampling one episode and updating the respective $v$ variables has $O(H)$ runtime. Theorem 1 states that after at most $\tilde{O}\left(\frac{H^2 C|\mathcal{S} \times \mathcal{A}|}{\epsilon^2} \ln \frac{1}{\delta}\right)$ observed episodes, the current policy is $\epsilon$-optimal with sufficiently high probability. This results in a total runtime for sampling of $\tilde{O}\left(\frac{H^3 C|\mathcal{S} \times \mathcal{A}|}{\epsilon^2} \ln \frac{1}{\delta}\right)$.

Each update of the policy involves updating the $n$ variables and $\mathcal{M}_k$ which takes runtime $O(C)$ and a call of FixedHorizonEVI with runtime cost $O(H|\mathcal{S}||\mathcal{A}|C + H|\mathcal{S}| \log |\mathcal{S}|)$. From Lemma C.1 below, we know that the policy can be updated at most $U_{\max}$ times which a gives total runtime for policy updates of

$$O(U_{\max} H|\mathcal{S}|(|\mathcal{A}|C + \log |\mathcal{S}|)) = O\left(H|\mathcal{S}|^2|\mathcal{A}|(|\mathcal{A}|C + \log |\mathcal{S}|) \log \frac{|\mathcal{S}|^2 H^2}{\epsilon}\right)$$

$$= \tilde{O}\left(H|\mathcal{S}|^2|\mathcal{A}|^2 C \log \frac{1}{\epsilon}\right).$$

The total runtime of UCFH before the policy is $\epsilon$-optimal with probability at least $1 - \delta$ is therefore

$$\tilde{O}\left(\frac{H^3|\mathcal{S}|^2|\mathcal{A}|^2 C}{\epsilon^2} \ln \frac{1}{\delta}\right).$$

The space complexity of UCFH is dominated by the requirement to store statistics for each possible transition which gives $O(|\mathcal{S}||\mathcal{A}|C)$ complexity.

# C    Detailed Proofs for the Upper PAC Bound

## C.1    Bound on the Number of Policy Changes of UCFH

**Lemma C.1.** *The total number of updates is bounded by* $U_{\max} = |\mathcal{S} \times \mathcal{A}| \log_2 \frac{|\mathcal{S}|H}{w_{\min}}$.

*Proof.* First note that $n(s, a)$ is never never decreasing and no updates happen once $n(s, a) \geq |\mathcal{S}|mH$ for all $(s, a)$. In each update, the $n(s, a)$ of exactly one $(s, a)$ pair increases by $\max\{mw_{\min}, n(s, a)\}$. For a single $(s, a)$ pair, such updates can happen only $\log_2(|\mathcal{S}|mH) - \log_2(mw_{\min})$ times. Hence, there are at most $|\mathcal{S} \times \mathcal{A}| \log_2 \frac{|\mathcal{S}|mH}{w_{\min}m}$ updates in total. □

## C.2    Proof of Lemma 1 – Capturing the true MDP

*Proof.* For a single $(s, a)$ pair, $s' \in \mathcal{S}(s, a)$ and $k$, we can treat the event that $s'$ is the successor state of $s$ when chosing action $a$ as a Bernoulli random variable with probability $p(s'|s, a)$. Using Hoeffding's inequality,[7] we then realize that

$$|p(s'|s, a) - \hat{p}(s'|s, a)| \leq \sqrt{\frac{\ln(6/\delta_1)}{2n}}$$

and by Bernstein's inequality

$$|p(s'|s, a) - \hat{p}(s'|s, a)| \leq \sqrt{\frac{2p(s'|s, a)(1 - p(s'|s, a))\ln(6/\delta_1)}{n}} + \frac{2}{3n}\ln(6/\delta_1)$$

with probability at least $1 - \delta_1/3$ respectively. Using both inequalities of Theorem 10 by Maurer and Pontil [18][8] and the fact that $(a + b)^2 \geq a^2 + b^2$, we have

$$|p(s'|s, a)(1 - p(s'|s, a)) - \tilde{p}(s'|s, a)(1 - \tilde{p}(s'|s, a))| \leq \frac{2\ln(6/\delta_1)}{n - 1}$$

for $n > 1$ with probability at least $1 - \delta_1/3$. All three inequalities hold with probability $1 - \delta_1$ by the union bound. By Lemma C.1, there are at most $U_{\max}$ updates and so there are at most $U_{\max}$ different $k$ to consider. Since in each update, only a single $(s, a)$ pair with at most $C$ successor states is updated, for all $k$ and $(s, a)$, there are only $U_{\max}C$ different $\hat{p}(s'|s, a)$ to consider. Applying the union bound, we get that $M \notin \mathcal{M}_k$ for any $k$ with probability at most $U_{\max}C\delta_1$. By setting $\delta_1 = \frac{\delta}{2CU_{\max}}$ we get the desired result. $\qquad\square$

## C.3  Bounding the number of episodes with $\kappa > |X_{k,\kappa,\iota}|$ for some $\kappa, \iota$

Before presenting the proof of Lemma 2 which bounds the total number of episodes where there is a $\kappa$ and $\iota$ such that $\kappa > |X_{k,\kappa,\iota}|$, we establish a bound for each individual $\kappa$ and $\iota$ in the following two additional lemmas.

**Lemma C.2** (Bound on observations of $X_{\cdot,\kappa,\iota}$)**.** *The total number of observations of $(s, a) \in X_{k,\kappa,\iota}$ where $\kappa \in [1, |\mathcal{S}| - 1]$ and $\iota > 0$ over all phases $k$ is at most $3|\mathcal{S} \times \mathcal{A}|mw_\iota\kappa$. The variable $w_\iota$ is the smallest possible weight of a $(s, a)$-pair that has importance $\iota$.*

*Proof.* We denote the smallest possible weight for any $(s, a)$ pair such that $\iota(s, a) = \iota$ by $w_\iota := \min\{w(s, a) : \iota_k(s, a) = \iota\}$. Note that $w_{\iota+1} = 2w_\iota$ for $\iota > 0$. Consider any phase $k$ and fix $(s, a) \in X_{k,\kappa,\iota}$ with $\iota > 0$. Since we assumed $\iota_k(s, a) = \iota > 0$, we have $w_\iota \leq w_k(s, a) < 2w_\iota$. From $\kappa_k(s, a) = \kappa$, it follows that

$$\frac{n_k(s, a)}{2mw_k(s, a)} \leq \kappa \leq \frac{n_k(s, a)}{mw_k(s, a)}$$

which implies that

$$mw_\iota\kappa \leq mw_k(s, a)\kappa \leq n_k(s, a) \leq 2mw_k(s, a)\kappa \leq 4mw_\iota\kappa. \qquad (4)$$

Hence, each state can only be observed $3mw_\iota$ times while being in $\{(s, a) \in X_{k,\kappa,\iota} : k \in \mathbb{N}\}$. $\quad\square$

**Lemma C.3.** *The number of episodes $E_{\kappa,\iota}$ in phases with $|X_{k,\kappa,\iota}| > \kappa$ is bounded for every $\alpha \geq 3$ with high probability,*

$$P(E_{\kappa,\iota} > \alpha N) \leq \exp\left(-\frac{\beta w_\iota(\kappa + 1)N}{H}\right)$$

*where $N = |\mathcal{S} \times \mathcal{A}|m$ and $\beta = \frac{\alpha(3/\alpha-1)^2}{7/3-1/\alpha}$.*

*Proof.* Let $\nu_i := \sum_{t=1}^{H} \mathbb{I}\{(s_t, a_t) \in X_{k,\kappa,\iota}\}$ be the number of observations of $(s, a)$ in $X_{k,\kappa,\iota}$ in the $i$th epsiode with $X_{k,\kappa,\iota} > \kappa$. We have $i \in \{1, \ldots E_{\kappa,\iota}\}$) and $k$ is the phase that episode $i$ belongs to.

Since $X_{k,\kappa,\iota} \geq \kappa + 1$ and all states in partition $(\kappa, \iota)$ have $w_k(s, a) \geq w_\iota$ , we get

$$\mathbb{E}[\nu_i|\nu_1, \ldots \nu_{i-1}] \geq (\kappa + 1)w_\iota. \qquad (5)$$

Also $\mathbb{V}[\nu_i|\nu_1 \ldots \nu_{i-1}] \leq \mathbb{E}[\nu_i|\nu_1, \ldots \nu_{i-1}]H$ as $\nu_i \in [0, H]$.

To reason about $E_{\kappa,\iota}$, we define the continuation

$$\nu_i^+ := \begin{cases} \nu_i & \text{if } i \leq E_{\kappa,\iota} \\ w_\iota(\kappa + 1) & \text{otherwise} \end{cases}$$

and the centered auxiliary sequence

$$\bar{\nu}_i := \frac{\nu_i^+ w_\iota(\kappa + 1)}{\mathbb{E}[\nu_i^+|\nu_1^+, \ldots \nu_{i-1}^+]}.$$

By construction

$$\mathbb{E}[\bar{\nu}_i|\bar{\nu}_1, \ldots \bar{\nu}_{i-1}] = w_\iota(\kappa + 1)\frac{\mathbb{E}[\nu_i^+|\bar{\nu}_1, \ldots, \bar{\nu}_{i-1}]}{\mathbb{E}[\nu_i^+|\nu_1^+, \ldots \nu_{i-1}^+]} = w_\iota(\kappa + 1).$$

By Lemma C.2, we have that $E_{\kappa,\iota} > \alpha N$ only if

$$\sum_{i=1}^{\alpha N} \bar{\nu}_i \leq 3Nw_\iota\kappa \leq 3Nw_\iota(\kappa+1).$$

Define now the martingale

$$B_i := \mathbb{E}\left[\sum_{j=1}^{\alpha N} \bar{\nu}_j | \bar{\nu}_1, \ldots \bar{\nu}_i\right] = \sum_{j=1}^{i} \bar{\nu}_j + \sum_{j=i+1}^{\alpha N} \mathbb{E}[\bar{\nu}_j | \bar{\nu}_1 \ldots \bar{\nu}_i]$$

which gives $B_0 = \alpha Nw_\iota(\kappa+1)$ and $B_{\alpha N} = \sum_{i=1}^{\alpha N} \bar{\nu}_i$. Further, since $\nu_i^+ \in [0, H]$ and Equation (5), we have

$$|B_{i+1} - B_i| = |\bar{\nu}_i - \mathbb{E}[\bar{\nu}_i | \bar{\nu}_1, \ldots, \bar{\nu}_{i-1}]| = \left|\frac{w_\iota(\kappa+1)(\nu_i^+ - \mathbb{E}[\nu_i^+ | \bar{\nu}_1, \ldots \bar{\nu}_{i-1}])}{\mathbb{E}[\nu_i^+ | \nu_1^+, \ldots \nu_{i-1}^+]}\right|$$

$$\leq \left|\nu_i^+ - \mathbb{E}[\nu_i^+ | \bar{\nu}_1, \ldots \bar{\nu}_{i-1}]\right| \leq H.$$

Using

$$\sigma^2 := \sum_{i=1}^{\alpha N} \mathbb{V}[B_i - B_{i-1} | B_1 - B_0, \ldots B_{i-1} - B_{i-2}]$$

$$= \sum_{i=1}^{\alpha N} \mathbb{V}[\bar{\nu}_i | \bar{\nu}_1, \ldots \bar{\nu}_{i-1}] \leq \alpha NHw_\iota(\kappa+1) = HB_0$$

we can apply Theorem 22 by Chung and Lu [21] and obtain

$$\mathbb{P}(E_{\kappa,\iota} > \alpha N) \leq \mathbb{P}\left(\sum_{i=1}^{\alpha N} \bar{\nu}_i \leq 3Nw_\iota(\kappa+1)\right)$$

$$= \mathbb{P}(B_{\alpha N} - B_0 \leq 3B_0/\alpha - B_0) = \mathbb{P}(B_{\alpha N} - B_0 \leq -(1 - 3/\alpha)B_0)$$

$$\leq \exp\left(-\frac{(3/\alpha - 1)^2 B_0^2}{2\sigma^2 + H(1/3 - 1/\alpha)B_0}\right)$$

for $\alpha \geq 3$. We can further simplify the bound to

$$\mathbb{P}(E_{\kappa,\iota} > \alpha N) \leq \exp\left(-\frac{(3/\alpha - 1)^2 B_0^2}{2HB_0 + H(1/3 - 1/\alpha)B_0}\right)$$

$$\leq \exp\left(-\frac{(3/\alpha - 1)^2}{2 + (-1/\alpha + 1/3)} \frac{B_0}{H}\right)$$

$$= \exp\left(-\frac{\alpha(3/\alpha - 1)^2}{7/3 - 1/\alpha} \frac{Nw_\iota(\kappa+1)}{H}\right).$$

$\square$

We are now ready to prove Lemma 2 by combining the bound in the previous lemma for all $\kappa$ and $\iota$.

**Proof of Lemma 2.** Since $w_k(s, a) \leq H$, we have that $\frac{w_k(s,a)}{w_{\min}} < \frac{H}{w_{\min}}$ and so $\iota_k(s, a) \leq H/w_{\min} = 4H^2|\mathcal{S}|/\epsilon$. In addition, $|X_{k,\kappa,\iota}| \leq |\mathcal{S}|$ for all $k, \kappa, \iota$ and so $|X_{k,\kappa,\iota}| > \kappa$ can only be true for $\kappa \leq |\mathcal{S}|$. Hence, only

$$E_{\max} = \log_2 \frac{H}{w_{\min}} \log_2 |\mathcal{S}|$$

possible values for $(\kappa, \iota)$ exists that can have $|X_{k,\kappa,\iota}| > \kappa$. Using the union bound over all $(\kappa, \iota)$ and Lemma C.3, we get that

$$\mathbb{P}(E \le \alpha N E_{\max}) \ge \mathbb{P}(\max_{(\kappa,\iota)} E_{\kappa,\iota} \le \alpha N) \ge 1 - E_{\max} \exp\left(-\frac{\beta w_\iota(\kappa+1)N}{H}\right)$$

$$\ge 1 - E_{\max} \exp\left(-\frac{\beta w_{\min}N}{H}\right) = 1 - E_{\max} \exp\left(-\frac{\beta w_{\min}m|\mathcal{S} \times \mathcal{A}|}{H}\right)$$

$$= 1 - E_{\max} \exp\left(-\frac{\beta \epsilon m|\mathcal{S} \times \mathcal{A}|}{4H^2|\mathcal{S}|}\right)$$

Bounding the right hand-side by $1 - \delta/2$ and solving for $m$ gives

$$1 - E_{\max} \exp\left(-\frac{\beta \epsilon m|\mathcal{S} \times \mathcal{A}|}{4H^2|\mathcal{S}|}\right) \ge 1 - \delta/2 \quad \Leftrightarrow \quad m \ge \frac{4H^2|\mathcal{S}|}{|\mathcal{S} \times \mathcal{A}|\beta\epsilon} \ln\frac{2E_{\max}}{\delta}$$

Hence, the condition

$$m \ge \frac{4H^2}{\beta\epsilon} \ln\frac{2E_{\max}}{\delta}$$

is sufficient for the desired result to hold. By plugging in $\alpha = 6$ and $\beta = \frac{\alpha(3/\alpha-1)^2}{7/3-1/\alpha} = \frac{9}{13} \ge \frac{2}{3}$, we obtain the statement to show. $\qquad \square$

### C.4  Bound on the value function difference for episodes with $\forall \kappa, \iota : |X_{k,\kappa,\iota}| \le \kappa$

To prove Lemma 3, it is sufficient to consider a fixed phase $k$. To avoid notational clutter, we therefore omit the phase indices $k$ in this section.

For the proof, we reason about a sequence of MDPs $M_d$ which have the same transition probabilities but different reward functions $r^{(d)}$. For $d = 0$, the reward function is the original reward function $r$ of $M$, i.e. $r_t^{(0)} = r_t$ for all $t = 1 \ldots H$. The following reward functions are then defined recursively as $r_t^{(2d+2)} = \sigma_{t:H}^{(d),2}$, where $\sigma_{t:H}^{(d),2}$ is the local variance of the value function w.r.t. the rewards $r^{(d)}$. Note that for every $d$ and $t = 1 \ldots H$ and $s \in \mathcal{S}$, we have $r_t^{(d)}(s) \in [0, H^d]$. In complete analogy, we define $\tilde{M}_d$ and $\hat{M}_d$.

We first prove a sequence of lemmas necessary for Lemma 3.

**Lemma C.4.**

$$V_{i,j} - \tilde{V}_{i,j} = \sum_{t=i}^{H-1} P_{i:t-1}(P_t - \tilde{P}_t)\tilde{V}_{t+1:j}$$

*Proof.*

$$V_{i,j}(s) - \tilde{V}_{i,j}(s) = r(s) + P_i V_{i+1:j}(s) - r(s) - \tilde{P}_i \tilde{V}_{i+1:j}(s) + P_i \tilde{V}_{i+1,j}(s) - P_i \tilde{V}_{i+1:j}(s)$$

$$= P_i(V_{i+1:j} - \tilde{V}_{i+1:j}) + (P_i - \tilde{P}_i)\tilde{V}_{i+1:j}(s)$$

Since we have $V_{j:j}(s) = r(s) = \tilde{V}_{j:j}(s)$, we can recursively expand the first difference until $i = j$ and get

$$V_{i,j} - \tilde{V}_{i,j} = \sum_{t=i}^{j-1} P_{i:t-1}(P_t - \tilde{P}_t)\tilde{V}_{t+1:j}$$

$\qquad \square$

**Lemma C.5.** *Assume $p, \hat{p}, \tilde{p} \in [0,1]$ satisfy $\hat{p} \in \mathcal{P}$ and $\tilde{p} \in \mathcal{P}$ where*

$$\mathcal{P} := \left\{ p' \in [0,1] : |p - p'| \le \sqrt{\frac{\ln(6/\delta_1)}{2n}}, \right.$$

$$|p - p'| \le \sqrt{\frac{2p(1-p)}{n}\ln(6/\delta_1)} + \frac{2}{3n}\ln(6/\delta_1),$$

$$\left. \text{if } n > 1 : |p'(1-p') - p(1-p)| \le \frac{2\ln(6/\delta_1)}{n-1} \right\}.$$

*Then*

$$|p - \tilde{p}| \leq \sqrt{\frac{8\tilde{p}(1-\tilde{p})}{n} \ln(6/\delta_1)} + \frac{16}{3(n-1)} \ln(6/\delta_1).$$

*Proof.*

$$|p - \tilde{p}| \leq |p - \hat{p}| + |\hat{p} - \tilde{p}| \leq 2\sqrt{\frac{2\hat{p}(1-\hat{p})}{n} \ln(6/\delta_1)} + 2\frac{2}{3n} \ln(6/\delta_1)$$

$$\leq 2\sqrt{\frac{2}{n}\left(\tilde{p}(1-\tilde{p}) + \frac{2\ln(6/\delta_1)}{n-1}\right)\ln(6/\delta_1)} + \frac{4}{3n}\ln(6/\delta_1)$$

$$\leq 2\sqrt{\frac{2\tilde{p}(1-\tilde{p})}{n}\ln(6/\delta_1)} + 2\frac{2\ln(6/\delta_1)}{n-1} + \frac{4}{3n}\ln(6/\delta_1)$$

$$\leq 2\sqrt{\frac{2\tilde{p}(1-\tilde{p})}{n}\ln(6/\delta_1)} + \frac{16}{3(n-1)}\ln(6/\delta_1)$$

$\square$

**Lemma C.6.** *Assume*
$$|p(s'|s,a) - \tilde{p}_i(s'|s,a)| \leq c_1(s,a) + c_2(s,a)\sqrt{\tilde{p}_i(s'|s,a)(1 - \tilde{p}_i(s'|s,a))}$$
*for $a = \pi_i(s)$ and all $s', s \in \mathcal{S}$. Then*
$$|(P_i - \tilde{P}_i)\tilde{V}_{i+1:j}(s)| \leq c_1(s,a)|\mathcal{S}(s,a)|\|\tilde{V}_{i+1:j}\|_\infty + c_2(s,a)\sqrt{|\mathcal{S}(s,a)|}\tilde{\sigma}_{i:j}(s)$$
*for any $(s,a) \in \mathcal{S} \times \mathcal{A}$ where $\mathcal{S}(s,a)$ denotes the set of possible successor states of state $s$ and action $a$.*

*Proof.* Let $s$ and $a = \pi_i(s)$ be fixed and define for this fixed $s$ the constant function $\bar{V}(s') = \tilde{P}_i\tilde{V}_{i+1:j}(s)$ *[sic]* as the expected value function of the successor states of $s$. Note that $\bar{V}(s')$ is a constant function and so $\bar{V} = \tilde{P}_i\bar{V} = P_i\bar{V}$.

$$|(P_i - \tilde{P}_i)\tilde{V}_{i+1:j}(s)| = |(P_i - \tilde{P}_i)\tilde{V}_{i+1:j}(s) + \bar{V}(s) - \bar{V}(s)|$$

$$= |(P_i - \tilde{P}_i)(\tilde{V}_{i+1:j} - \bar{V})(s)|$$

$$\leq \sum_{s' \in \mathcal{S}(s,a)} |p(s'|s,a) - \tilde{p}_i(s'|s,a)||\tilde{V}_{i+1:j}(s') - \bar{V}(s')| \quad (6)$$

$$\leq \sum_{s' \in \mathcal{S}(s,a)} \left(c_1(s,a) + c_2(s,a)\sqrt{\tilde{p}_i(s'|s,a)(1 - \tilde{p}_i(s'|s,a))}\right)|\tilde{V}_{i+1:j}(s') - \bar{V}(s')|$$

$$\leq |\mathcal{S}(s,a)|c_1(s,a)\|\tilde{V}_{i+1:j}\|_\infty + c_2(s,a)\sum_{s' \in \mathcal{S}(s,a)}\sqrt{\tilde{p}_i(s'|s,a)(1 - \tilde{p}_i(s'|s,a))(\tilde{V}_{i+1:j}(s') - \bar{V}(s'))^2}$$

$$\leq |\mathcal{S}(s,a)|c_1(s,a)\|\tilde{V}_{i+1:j}\|_\infty + c_2(s,a)\sqrt{|\mathcal{S}(s,a)|\sum_{s' \in \mathcal{S}(s,a)}\tilde{p}_i(s'|s,a)(1 - \tilde{p}_i(s'|s,a))(\tilde{V}_{i+1:j}(s') - \bar{V}(s'))^2}$$

$$(7)$$

$$\leq |\mathcal{S}(s,a)|c_1(s,a)\|\tilde{V}_{i+1:j}\|_\infty + c_2(s,a)\sqrt{|\mathcal{S}(s,a)|\sum_{s' \in \mathcal{S}(s,a)}\tilde{p}_i(s'|s,a)(\tilde{V}_{i+1:j}(s') - \bar{V}(s'))^2}$$

$$= |\mathcal{S}(s,a)|c_1(s,a)\|\tilde{V}_{i+1:j}\|_\infty + c_2(s,a)\sqrt{|\mathcal{S}(s,a)|}\tilde{\sigma}_{i:j}(s)$$

In Inequality (6), we wrote out the definition of $P_i$ and $\tilde{P}_i$ and applied the triangle inequality. We then applied the assumed bound and bounded $|\tilde{V}_{i+1:j}(s') - \bar{V}(s')|$ by $\|V_{i+1:j}\|_\infty$ as all value functions are nonnegative. In Inequality (7), we applied the Cauchy-Schwarz inequality and subsequently used the fact that each term is the sum is nonnegative and that $(1 - \tilde{p}_i(s'|s,a)) \leq 1$. The final equality follows from the definition of $\tilde{\sigma}_{i:j}$. $\square$

### C.4.1 Bounding the difference in value function

**Lemma C.7.** *Assume $M \in \mathcal{M}_k$. If $|X_{\kappa,\iota}| \leq \kappa$ for all $(\kappa, \iota)$. Then*

$$|V_{1:H}^{(d)}(s_0) - \tilde{V}_{1:H}^{(d)}(s_0)| =: \Delta_d \leq \hat{A}_d + \hat{B}_d + \min\{\hat{C}_d, \hat{C}'_d + \hat{C}'' \sqrt{\Delta_{2d+2}}\}$$

*where*

$$\hat{A}_d = \frac{\epsilon}{4} H^d, \qquad \hat{B}_d = \frac{32 H^{d+1} |\mathcal{K} \times \mathcal{I}| C}{3m} \ln \frac{6}{\delta_1},$$

*and*

$$\hat{C}'_d = \sqrt{C |\mathcal{K} \times \mathcal{I}| \frac{8}{m} H^{2d+2} \ln \frac{6}{\delta_1}} \qquad \hat{C}_d = \hat{C}'_d \sqrt{H}, \qquad \hat{C}'' = \sqrt{C |\mathcal{K} \times \mathcal{I}| \frac{8}{m} \ln \frac{6}{\delta_1}}.$$

*Proof.*

$$\Delta_d = |V_{1:H}^{(d)}(s_0) - \tilde{V}_{1:H}^{(d)}(s_0)| = \left| \sum_{t=1}^{H-1} P_{1:t-1}(P_t - \tilde{P}_t) \tilde{V}_{t+1:H}^{(d)}(s_0) \right|$$

$$\leq \sum_{t=1}^{H-1} P_{1:t-1} |(P_t - \tilde{P}_t) \tilde{V}_{t+1:H}^{(d)}|(s_0)$$

$$= \sum_{t=1}^{H-1} P_{1:t-1} \left( \sum_{s,a \in \mathcal{S} \times \mathcal{A}} \mathbb{I}\{s = \cdot, a = \pi_t(s)\} |(P_t - \tilde{P}_t) \tilde{V}_{t+1:H}^{(d)}| \right)(s_0)$$

$$= \sum_{s,a \in \mathcal{S} \times \mathcal{A}} \sum_{t=1}^{H-1} P_{1:t-1} \left( \mathbb{I}\{s = \cdot, a = \pi_t(s)\} |(P_t - \tilde{P}_t) \tilde{V}_{t+1:H}^{(d)}| \right)(s_0)$$

$$= \sum_{s,a \in \mathcal{S} \times \mathcal{A}} \sum_{t=1}^{H-1} P_{1:t-1} \left( \mathbb{I}\{s = \cdot, a = \pi_t(s)\} |(P_t - \tilde{P}_t) \tilde{V}_{t+1:H}^{(d)}(s)| \right)(s_0)$$

The first equality follows from Lemma C.4, the second step from the fact that $V_{t+1:H} \geq 0$ and $P_{1:t-1}$ being non-expansive. In the third, we introduce an indicator function which does not change the value as we sum over all $(s, a)$ pairs. The fourth step relies on the linearity of the $P_{i:j}$ operators. In the fifth step, we realize that $\mathbb{I}\{s = \cdot, a = \pi_t(s)\} |(P_t - \tilde{P}_t) \tilde{V}_{t+1:H}^{(d)}(\cdot)|$ is a function that takes nonzero values only for input $s$. We can therefore replace the argument of the second term with $s$ without changing the value. The term then becomes constant and by linearity of $P_{i:j}$, we can write

$$|V_{1:H}^{(d)}(s_0) - \tilde{V}_{1:H}^{(d)}(s_0)| = \Delta_d \leq \sum_{s,a \in \mathcal{S} \times \mathcal{A}} \sum_{t=1}^{H-1} |(P_t - \tilde{P}_t) \tilde{V}_{t+1:H}^{(d)}(s)| (P_{1:t-1} \mathbb{I}\{s = \cdot, a = \pi_t(s)\})(s_0)$$

$$\leq \sum_{s,a \notin X} \sum_{t=1}^{H-1} \|\tilde{V}_{t+1:H}^{(d)}\|_\infty (P_{1:t-1} \mathbb{I}\{s = \cdot, a = \pi_t(s)\})(s_0)$$

$$+ \sum_{s,a \in X} \sum_{t=1}^{H-1} |(P_t - \tilde{P}_t) \tilde{V}_{t+1:H}^{(d)}(s)| (P_{1:t-1} \mathbb{I}\{s = \cdot, a = \pi_t(s)\})(s_0)$$

$$\leq \sum_{s,a \notin X} \sum_{t=1}^{H-1} H^{d+1} (P_{1:t-1} \mathbb{I}\{s = \cdot, a = \pi_t(s)\})(s_0)$$

$$+ \sum_{s,a \in X} \sum_{t=1}^{H-1} |(P_t - \tilde{P}_t) \tilde{V}_{t+1:H}^{(d)}(s)| (P_{1:t-1} \mathbb{I}\{s = \cdot, a = \pi_t(s)\})(s_0)$$

$$\leq \sum_{s,a \notin X} \sum_{t=1}^{H-1} H^{d+1}(P_{1:t-1}\mathbb{I}\{s = \cdot, a = \pi_t(s)\})(s_0)$$

$$+ \sum_{s,a \in X} \sum_{t=1}^{H-1} \left| |\mathcal{S}(s,a)|c_1(s,a)H^{d+1} + c_2(s,a)\sqrt{|\mathcal{S}(s,a)|}\tilde{\sigma}_{t:H}^{(d)}(s,a) \right| (P_{1:t-1}\mathbb{I}\{s = \cdot, a = \pi_t(s)\})(s_0)$$

$$\leq \sum_{s,a \notin X} \sum_{t=1}^{H} H^{d+1}(P_{1:t-1}\mathbb{I}\{s = \cdot, a = \pi_t(s)\})(s_0)$$

$$+ \sum_{s,a \in X} \sum_{t=1}^{H} \left| |\mathcal{S}(s,a)|c_1(s,a)H^{d+1} \right| (P_{1:t-1}\mathbb{I}\{s = \cdot, a = \pi_t(s)\})(s_0)$$

$$+ \sum_{s,a \in X} \sum_{t=1}^{H-1} \left| c_2(s,a)\sqrt{|\mathcal{S}(s,a)|}\tilde{\sigma}_{t:H}^{(d)}(s,a) \right| (P_{1:t-1}\mathbb{I}\{s = \cdot, a = \pi_t(s)\})(s_0)$$

$$\leq \sum_{s,a \notin X} H^{d+1}w(s,a) + \sum_{s,a \in X} |\mathcal{S}(s,a)|c_1(s,a)H^{d+1}w(s,a)$$

$$+ \sum_{s,a \in X} \sqrt{|\mathcal{S}(s,a)|}c_2(s,a) \sum_{t=1}^{H-1} \tilde{\sigma}_{t:H}^{(d)}(s,a)(P_{1:t-1}\mathbb{I}\{s = \cdot, a = \pi_t(s)\})(s_0)$$

$$\leq \sum_{s,a \notin X} H^{d+1}w(s,a) + \sum_{s,a \in X} Cc_1(s,a)H^{d+1}w(s,a)$$

$$+ \sum_{s,a \in X} \sqrt{C}c_2(s,a) \sum_{t=1}^{H-1} \tilde{\sigma}_{t:H}^{(d)}(s,a)(P_{1:t-1}\mathbb{I}\{s = \cdot, a = \pi_t(s)\})(s_0)$$

In the second inequality, we split the sum over all $(s,a)$ pairs and used the fact that $P_t$ and $\tilde{P}_t$ are non-expansive, i.e., $|(P_t - \tilde{P}_t)\tilde{V}_{t+1:H}^{(d)}(s)| \leq \|V_{t+1:H}^{(d)}\|_\infty$. The next step follows from $\|V_{t+1:H}^{(d)}\|_\infty \leq \|V_{1:H}^{(d)}\|_\infty \leq H^{d+1}$. We then apply Lemma C.6 and subsequently use that all terms are nonnegative and the definition of $w(s,a)$. Eventually, we use that $|\mathcal{S}(s,a)| \leq C$ for all $s,a$. Using the assumption that $M \in \mathcal{M}_k$ and $\tilde{M} \in \mathcal{M}_k$ from Lemma A.1, we can apply Lemma C.5 and get that

$$c_2(s,a) = \sqrt{\frac{8}{n(s,a)} \ln \frac{6}{\delta_1}} \quad \text{and} \quad c_1(s,a) = \frac{16}{3(n(s,a)-1)} \ln \frac{6}{\delta_1}.$$

Hence, we can bound

$$\Delta_d \leq A(s_0) + B(s_0) + C(s_0)$$

as a sum of three terms which we will consider individually in the following. The first term is

$$A(s_0) = \sum_{s,a \notin X} H^{d+1}w(s,a) \leq w_{\min}|\mathcal{S}|H^{d+1} \leq \frac{\epsilon H^{d+1}|\mathcal{S}|}{4H|\mathcal{S}|} = \frac{\epsilon}{4}H^d = \hat{A}_d$$

as $w(s,a) \leq w_{\min}$ for all $s,a$ not in the active set and that the policy is deterministic, which implies that there are only $|\mathcal{S}|$ nonzero $w$. The next term is

$$B(s_0) = C \sum_{s,a \in X} w(s,a)H^{d+1}\frac{16}{3(n(s,a)-1)} \ln \frac{6}{\delta_1}$$

$$= H^{d+1}C \ln \frac{6}{\delta_1} \sum_{\kappa,\iota} \sum_{s,a \in X_{\kappa,\iota}} w(s,a)\frac{16}{3(n(s,a)-1)}$$

$$\leq H^{d+1}\frac{16C}{3} \ln \frac{6}{\delta_1} \sum_{\kappa,\iota} \sum_{s,a \in X_{\kappa,\iota}} \frac{w(s,a)}{n(s,a)} \frac{n(s,a)}{n(s,a)-1}.$$

For $s, a \in X_{\kappa, \iota}$, we have $n(s, a) \geq m w(s, a) \kappa$ (see Equation (4)) and so

$$\frac{w(s, a)}{n(s, a)} \leq \frac{1}{\kappa m}. \tag{8}$$

Further, for all relevant $(s, a)$-pairs, we have $n(s, a) > 1$ (follows from $|X_{\kappa, \iota}| \leq \kappa$) which implies

$$B(s_0) \leq H^{d+1} \frac{32 C}{3} \ln \frac{6}{\delta_1} \sum_{\kappa, \iota} \frac{|X_{\kappa, \iota}|}{\kappa m}$$

and since we assumed $|X_{\kappa, \iota}| \leq \kappa$

$$B(s_0) \leq \frac{32 H^{d+1} |\mathcal{K} \times \mathcal{I}| C}{3m} \ln \frac{6}{\delta_1} = \hat{B}_d$$

where $\mathcal{K} \times \mathcal{I}$ is the set of all possible $(\kappa, \iota)$-pairs. The last term is

$$C(s_0) = \sqrt{C} \sum_{s, a \in X} c_2(s, a) \sum_{t=1}^{H-1} \tilde{\sigma}_{t:H}^{(d)}(s, a)) P_{1:t-1} \mathbb{I}\{s = \cdot, a = \pi_t(s)\}$$

$$\leq \sqrt{C} \sum_{s, a \in X} c_2(s, a) \sum_{t=1}^{H-1} \tilde{\sigma}_{t:H}^{(d)}(s, a)) P_{1:t-1} \mathbb{I}\{s = \cdot, a = \pi_t(s)\}$$

$$\leq \sqrt{C} \sum_{s, a \in X} c_2(s, a) \sqrt{\sum_{t=1}^{H-1} P_{1:t-1} \mathbb{I}\{s = \cdot, a = \pi_t(s)\}} \sqrt{\sum_{t=1}^{H-1} \tilde{\sigma}_{t:H}^{(d),2}(s, a)) P_{1:t-1} \mathbb{I}\{s = \cdot, a = \pi_t(s)\}}$$

$$\leq \sqrt{C} \sum_{s, a \in X} \sqrt{\frac{8 w(s, a)}{n(s, a)} \ln \frac{6}{\delta_1} \sum_{t=1}^{H-1} \tilde{\sigma}_{t:H}^{(d),2}(s, a)) P_{1:t-1} \mathbb{I}\{s = \cdot, a = \pi_t(s)\}}$$

where we first applied the Cauchy-Schwarz inequality and then used the definition of $c_2(s, a)$ and $w(s, a)$.

$$C(s_0) \leq \sqrt{C} \sum_{\kappa, \iota} \sum_{s, a \in X_{\kappa, \iota}} \sqrt{\frac{8 w(s, a)}{n(s, a)} \ln \frac{6}{\delta_1} \sum_{t=1}^{H-1} \tilde{\sigma}_{t:H}^{(d),2}(s, a)) P_{1:t-1} \mathbb{I}\{s = \cdot, a = \pi_t(s)\}(s_0)}$$

$$\leq \sqrt{C} \sum_{\kappa, \iota} \sqrt{|X_{\kappa, \iota}| \sum_{s, a \in X_{\kappa, \iota}} \frac{8 w(s, a)}{n(s, a)} \ln \frac{6}{\delta_1} \sum_{t=1}^{H-1} \tilde{\sigma}_{t:H}^{(d),2}(s, a)) P_{1:t-1} \mathbb{I}\{s = \cdot, a = \pi_t(s)\}(s_0)}$$

$$\leq \sqrt{C} \sum_{\kappa, \iota} \sqrt{\sum_{s, a \in X_{\kappa, \iota}} \frac{8}{m} \ln \frac{6}{\delta_1} \sum_{t=1}^{H-1} \tilde{\sigma}_{t:H}^{(d),2}(s, a)) P_{1:t-1} \mathbb{I}\{s = \cdot, a = \pi_t(s)\}(s_0)}$$

$$\leq \sqrt{C |\mathcal{K} \times \mathcal{I}| \frac{8}{m} \ln \frac{6}{\delta_1} \sum_{s, a \in X} \sum_{t=1}^{H-1} \tilde{\sigma}_{t:H}^{(d),2}(s, a)) P_{1:t-1} \mathbb{I}\{s = \cdot, a = \pi_t(s)\}(s_0)}$$

$$\leq \sqrt{C |\mathcal{K} \times \mathcal{I}| \frac{8}{m} \ln \frac{6}{\delta_1} \sum_{s, a \in \mathcal{S} \times \mathcal{A}} \sum_{t=1}^{H-1} \tilde{\sigma}_{t:H}^{(d),2}(s, a)) P_{1:t-1} \mathbb{I}\{s = \cdot, a = \pi_t(s)\}(s_0)}$$

$$= \sqrt{C |\mathcal{K} \times \mathcal{I}| \frac{8}{m} \ln \frac{6}{\delta_1} \sum_{t=1}^{H-1} P_{1:t-1} \tilde{\sigma}_{t:H}^{(d),2}(s_0)} \tag{9}$$

$$\leq \sqrt{C |\mathcal{K} \times \mathcal{I}| \frac{8 H^{2d+3} \ln(6/\delta_1)}{m}} = \hat{C}_d$$

We first split the sum and applied the Cauchy-Schwarz inequality. Then we used again Inequality (8) and $|X_{\kappa,\iota}| \leq \kappa$. In the fourth step, we applied Cauchy-Schwarz and the final inequality follows from $\|\tilde{\sigma}_{t:H}^{(d),2}\|_\infty \leq H^{2d+2}$ and the fact that $P_{1:t-1}$ is non-expansive. Alternatively, we can rewrite the bound in Equation (9) as

$$
C(s_0) \leq \sqrt{C\,|\mathcal{K}\times\mathcal{I}|\,\frac{8}{m}\ln\frac{6}{\delta_1}\sum_{t=1}^{H-1}P_{1:t-1}\tilde{\sigma}_{t:H}^{(d),2}(s_0)}
$$

$$
= \sqrt{C\,|\mathcal{K}\times\mathcal{I}|\,\frac{8}{m}\ln\frac{6}{\delta_1}\sum_{t=1}^{H-1}P_{1:t-1}\tilde{\sigma}_{t:H}^{(d),2}(s_0) - \tilde{P}_{1:t-1}\tilde{\sigma}_{t:H}^{(d),2}(s_0) + \tilde{P}_{1:t-1}\tilde{\sigma}_{t:H}^{(d),2}(s_0)}.
$$

Lemma 4 shows that the variance $\tilde{\mathcal{V}}_{1:H}^{(d)}$ also satisfies the Bellman equation with the local variances $\tilde{\sigma}_{i:j}^{(d),2}$. This insight allows us to bound $\sum_{t=1}^{H-1}\tilde{P}_{1:t-1}\tilde{\sigma}_{t:H}^{(d),2}(s_0) = \tilde{\mathcal{V}}_{1:H}^{(d)}(s_0) \leq H^{2d+2}$. Also, note that $\tilde{\sigma}_{t:H}^{(d),2} = r_t^{(2d+2)}$ which gives us

$$
C(s_0) \leq \sqrt{C\,|\mathcal{K}\times\mathcal{I}|\,\frac{8}{m}\ln\frac{6}{\delta_1}\left(H^{2d+2} + \sum_{t=1}^{H-1}P_{1:t-1}r_t^{(2d+2)}(s_0) - \tilde{P}_{1:t-1}r_t^{(2d+2)}(s_0)\right)}
$$

$$
= \sqrt{C\,|\mathcal{K}\times\mathcal{I}|\,\frac{8}{m}\ln\frac{6}{\delta_1}\left(H^{2d+2} + V_{1:H}^{(2d+2)}(s_0) - \tilde{V}_{1:H}^{(2d+2)}(s_0)\right)}
$$

$$
\leq \sqrt{C\,|\mathcal{K}\times\mathcal{I}|\,\frac{8}{m}\ln\frac{6}{\delta_1}\left(H^{2d+2} + \Delta_{2d+2}\right)}
$$

$$
\leq \sqrt{C\,|\mathcal{K}\times\mathcal{I}|\,\frac{8}{m}H^{2d+2}\ln\frac{6}{\delta_1}} + \sqrt{C\,|\mathcal{K}\times\mathcal{I}|\,\frac{8}{m}\Delta_{2d+2}\ln\frac{6}{\delta_1}} = \hat{C}_d' + \hat{C}''\sqrt{\Delta_{2d+2}}
$$

$\square$

### C.4.2 Proof of Lemma 4 (Bellman equation of local value function variances)

*Proof of Lemma 4.*

$$
\mathcal{V}_{i:j}(s) = \mathbb{E}\left[\left(\sum_{t=i}^{j}r_t(s_t) - V_{i:j}(s_i)\right)^2 \Big| s_i = s\right]
$$

$$
= \mathbb{E}\left[\left(\sum_{t=i+1}^{j}r_t(s_t) - V_{i+1:j}(s_{i+1}) + V_{i+1:j}(s_{i+1}) + r_i(s_i) - V_{i:j}(s_i)\right)^2 \Big| s_i = s\right]
$$

$$
= \mathbb{E}\left[\left(\sum_{t=i+1}^{j}r(s_t) - V_{i+1:j}(s_{i+1})\right)^2 \Big| s_i = s\right]
$$

$$
+ 2\mathbb{E}\left[\left(\sum_{t=i+1}^{j}r_t(s_t) - V_{i+1:j}(s_{i+1})\right)(V_{i+1:j}(s_{i+1}) + r_i(s_i) - V(s_i))\,|s_i = s\right]
$$

$$
+ \mathbb{E}\left[(V_{i+1:j}(s_{i+1}) + r_i(s_i) - V_{i:j}(s_i))^2 \,|s_i = s\right]
$$

$$
= \mathbb{E}\left[\mathcal{V}_{i+1:j}(s_{i+1})|s_i = s\right]
$$

$$
+ 2\mathbb{E}\left[\mathbb{E}\left[\left(\sum_{t=i+1}^{j}r_t(s_t) - V_{i+1:j}(s_{i+1})\right)(V_{i+1:j}(s_{i+1}) + r_i(s_i) - V_{i:j}(s_i))\,|s_{i+1}\right] \Big| s_i = s\right]
$$

$$
+ \mathbb{E}\left[(V_{i+1:j}(s_{i+1}) - P_i V_{i+1:j}(s_i))^2 \,|s_i = s\right]
$$

where the final equality follows from the tower property of conditional expectations, and the fact that $V_{i:j}(s_i) = P_i V_{i+1:j}(s_i) + r_i(s_i)$. Since by the definition of the value function

$$\mathbb{E}\left[\left(\sum_{t=i+1}^{j} r_t(s_t) - V_{i+1:j}(s_{i+1})\right)|s_{i+1}\right] = 0$$

the middle term vanishes and the last term is by definition $\sigma_{i:j}^2(s)$ we obtain

$$\mathcal{V}_{i:j}(s) = P_i \mathcal{V}_{i+1:j}(s) + \sigma_{i:j}^2(s).$$

Noting that $\mathcal{V}_{j:j}(s) = (r_j(s) - r_j(s))^2 = 0$, we can unroll the equation and obtain

$$\mathcal{V}_{i:j}(s) = \sum_{t=i}^{j} P_{i:t-1}\sigma_{t:j}^2(s).$$

From the definition of $\mathcal{V}_{1:H}$ and the fact that $0 \le r(\cdot) \le r_{\max}$, we see that $0 \le \mathcal{V}_{1:H} \le H^2 r_{\max}^2$ and the final statement of the lemma follows.

$\square$

### C.4.3 Proof of Lemma 3

*Proof of Lemma 3.* The recursive bound from Lemma C.7

$$\Delta_d \le \hat{A}_d + \hat{B}_d + \hat{C}_d' + \hat{C}''\sqrt{\Delta_{2d+2}}$$

has the form $\Delta_d \le Y_d + Z\sqrt{\Delta_{2d+2}}$. Expanding this form and using the triangle inequality gives

$$\Delta_0 \le Y_0 + Z\sqrt{\Delta_2} \le Y_0 + Z\sqrt{Y_2 + Z\sqrt{\Delta_6}} \le Y_0 + Z\sqrt{Y_2} + Z^{3/2}\Delta_6^{1/4}$$
$$\le Y_0 + Z\sqrt{Y_2} + Z^{3/2}Y_6^{1/4} + Z^{7/4}\Delta_{14}^{1/8} \le \ldots$$

and by doing this up to level $\gamma = \lceil\frac{\ln H}{2\ln 2}\rceil$, we obtain

$$\Delta_0 \le \sum_{d\in\mathcal{D}\setminus\{\gamma\}} Z^{\frac{2d}{2+d}}Y_d^{\frac{2}{2+d}} + Z^{\frac{2\gamma}{2+\gamma}}\Delta_\gamma^{\frac{2}{2+\gamma}}$$

where $\mathcal{D} = \{0, 2, 6, 14, \ldots \gamma\}$. Note that the exponent of $H$ compared to $m$ is the larger in $\hat{C}_d'$ than in $\hat{B}_d$. Therefore, for sufficiently large $m$, $\hat{C}_d'$ dominates the other term. More precisely, for

$$m \ge \frac{128H}{9}C|\mathcal{K} \times \mathcal{I}| \ln\frac{6}{\delta_1} \tag{10}$$

we have $\hat{B}_d \le \hat{C}_d'$. We can therefore consider $Z = \hat{C}''$ and $Y_d = 2\hat{C}_d' + \hat{A}_d$. Also, since $\hat{C}_d \ge \hat{C}_d'$, we can bound $\Delta_\gamma \le \hat{A}_d + 2\hat{C}_d$. For notational simplicity, we will use the auxiliary variable

$$m_1 = \frac{8C|\mathcal{K} \times \mathcal{I}|H^2}{m\epsilon^2}\ln\frac{6}{\delta_1}.$$

and get

$$Z = \hat{C}'' = \sqrt{m_1}\frac{\epsilon}{H} \quad\text{and}$$
$$Y_d = \hat{A}_d + 2\hat{C}_d' = (1/4 + 2\sqrt{m_1})H^d\epsilon \quad\text{and}$$
$$\Delta_\gamma \le \hat{A}_\gamma + 2\hat{C}_\gamma = (1/4 + 2\sqrt{m_1 H})H^\gamma\epsilon.$$

Then

$$\left(Z^{2d}Y_d^2\right)^{(2+d)^{-1}} = \left(m_1^d\epsilon^{2d+2}(1/4 + 2\sqrt{m_1})^2\right)^{(2+d)^{-1}} = \epsilon\left(m_1^d\epsilon^d(1/4 + 2\sqrt{m_1})^2\right)^{(2+d)^{-1}}$$

and

$$\left(Z^{2\gamma}\Delta_\gamma\right)^{(2+\gamma)^{-1}} = \left(m_1^\gamma \epsilon^{2\gamma+2}(1/4 + 2\sqrt{m_1 H})^2\right)^{(2+\gamma)^{-1}} = \epsilon \left(m_1^\gamma \epsilon^\gamma (1/4 + 2\sqrt{m_1 H})^2\right)^{(2+\gamma)^{-1}}.$$

Putting these pieces together, we obtain

$$\begin{aligned}
\frac{\Delta_0}{\epsilon} &\leq \sum_{d \in \mathcal{D}\setminus\{\gamma\}} (\epsilon m_1)^{\frac{d}{2+d}} \left(\frac{1}{4} + 2\sqrt{m_1}\right)^{\frac{2}{d+2}} + (\epsilon m_1)^{\frac{\gamma}{\gamma+2}} \left(\frac{1}{4} + 2\sqrt{Hm_1}\right)^{\frac{2}{\gamma+2}} \\
&= \frac{1}{4} + 2\sqrt{m_1} + \sum_{d \in \mathcal{D}\setminus\{0,\gamma\}} (\epsilon m_1)^{\frac{d}{2+d}} \left(\frac{1}{4} + 2\sqrt{m_1}\right)^{\frac{2}{d+2}} + (\epsilon m_1)^{\frac{\gamma}{\gamma+2}} \left(\frac{1}{4} + 2\sqrt{Hm_1}\right)^{\frac{2}{\gamma+2}} \\
&\leq \frac{1}{4} + 2\sqrt{m_1} + \sum_{d \in \mathcal{D}\setminus\{0,\gamma\}} (\epsilon m_1)^{\frac{d}{2+d}} \left[\left(\frac{1}{4}\right)^{\frac{2}{d+2}} + (2\sqrt{m_1})^{\frac{2}{d+2}}\right] \\
&\quad + (\epsilon m_1)^{\frac{\gamma}{\gamma+2}} \left[\left(\frac{1}{4}\right)^{\frac{2}{\gamma+2}} + \left(2\sqrt{Hm_1}\right)^{\frac{2}{\gamma+2}}\right]
\end{aligned}$$

where we used the fact that $(a+b)^\phi \leq a^\phi + b^\phi$ for $a, b > 0$ and $0 < \phi < 1$. We now bound the $H^{1/(2+\gamma)}$ by using the definition of $\gamma$. Since

$$\frac{1}{2+\gamma} = \frac{2\ln 2}{4\ln 2 + \ln H} \leq 2\log_H 2$$

and since $H \geq 1$, we have $H^{1/(2+\gamma)} \leq 4$. Therefore

$$\begin{aligned}
\frac{\Delta_0}{\epsilon} &\leq \frac{1}{4} + 2\sqrt{m_1} + \sum_{d \in \mathcal{D}\setminus\{0,\gamma\}} (\epsilon m_1)^{\frac{d}{2+d}} \left[\left(\frac{1}{4}\right)^{\frac{2}{d+2}} + (2\sqrt{m_1})^{\frac{2}{d+2}}\right] \\
&\quad + (\epsilon m_1)^{\frac{\gamma}{\gamma+2}} \left[\left(\frac{1}{4}\right)^{\frac{2}{\gamma+2}} + 4\left(2\sqrt{m_1}\right)^{\frac{2}{\gamma+2}}\right] \\
&\leq \frac{1}{4} + 2\sqrt{m_1} + \sum_{d \in \mathcal{D}\setminus\{0\}} (\epsilon m_1)^{\frac{d}{2+d}} \left[\left(\frac{1}{4}\right)^{\frac{2}{d+2}} + 4\left(2\sqrt{m_1}\right)^{\frac{2}{d+2}}\right] \\
&\leq \frac{1}{4} + 2\sqrt{m_1} + \sum_{i=1}^{\log_2 \gamma} (\epsilon m_1)^{1-2^{-i}} \left[\left(\frac{1}{4}\right)^{2^{-i}} + 4\left(2\sqrt{m_1}\right)^{2^{-i}}\right] \\
&\leq \frac{1}{4} + 2\sqrt{m_1} + \sum_{i=1}^{\log_2 \gamma} m_1^{1-2^{-i}} \left[\left(\frac{1}{4}\right)^{2^{-i}} + 4\left(2\sqrt{m_1}\right)^{2^{-i}}\right]
\end{aligned}$$

In the first inequality, we used the bound for $H^{1/(2+\gamma)}$ and in the second inequality we simplified the expression by noting that all terms are nonnegative. In the next step, we re-parameterized the sum. In the final inequality, we used the assumption that $0 < \epsilon \leq 1$ and therefore $\epsilon^{1-2^{-i}} \leq 1$.

$$\begin{aligned}
\frac{\Delta_0}{\epsilon} &\leq \frac{1}{4} + 2\sqrt{m_1} + \frac{1}{4}\sum_{i=1}^{\log_2 \gamma} (4m_1)^{1-2^{-i}} + 4\sum_{i=1}^{\log_2 \gamma} (m_1)^{1-2^{-i}} (4m_1)^{2^{-i-1}} \\
&\leq \frac{1}{4} + 2\sqrt{m_1} + \frac{1}{4}\sum_{i=1}^{\log_2 \gamma} (4m_1)^{1-2^{-i}} + 16\sum_{i=1}^{\log_2 \gamma} \left(\frac{m_1}{4}\right)^{1-2^{-i-1}}.
\end{aligned}$$

By requiring that

$$m_1 \leq \frac{1}{4}$$

and noting that $1 - 2^{-i} \geq 1/2$ and $1 - 2^{-i-1} \geq 3/4$ for $i \geq 1$, we can bound the expression by

$$\frac{\Delta_0}{\epsilon} \leq \frac{1}{4} + 2\sqrt{m_1} + \frac{1}{4}\log_2(\gamma)\sqrt{4m_1} + 16\log_2(\gamma)\left(\frac{m_1}{4}\right)^{3/4}.$$

By requiring that $m_1 \leq 1/64$ and $m_1 \leq (2\log_2 \gamma)^{-2}$ and $m_1 \leq 1/64(\log_2 \gamma)^{-4/3}$, we can assure that $\Delta_0 \leq \epsilon$. Taking all assumptions on $m_1$ we made above together, we realize that

$$m_1 \leq \left(\frac{1}{8\log_2 \log_2 H}\right)^2 \leq \left(\frac{1}{8\log_2 \gamma}\right)^2$$

is sufficient for them to hold where we used $\log_2 \gamma = \log_2(\lceil \frac{1}{2}\log_2 H\rceil) \leq \log_2 \log_2 H$. This gives the following condition on $m$

$$m \geq 512C(\log_2 \log_2 H)^2 |\mathcal{K} \times \mathcal{I}|\frac{H^2}{\epsilon^2}\ln\frac{6}{\delta_1}$$

which is a stronger condition that the one in Equation (10).

By construction of $\iota(s,a)$, we have $\iota(s,a) \leq 2\frac{H}{w_{\min}} = \frac{8|\mathcal{S}|H^2}{\epsilon} = \frac{8H^2|\mathcal{S}|}{\epsilon}$. Also, $\kappa_k(s,a) \leq \frac{|\mathcal{S}|mH}{mw_{\min}} = \frac{4|\mathcal{S}|^2H^2}{\epsilon}$. Therefore

$$|\mathcal{K} \times \mathcal{I}| \leq \log_2 \frac{4|\mathcal{S}|^2H^2}{\epsilon}\log_2\frac{8H^2|\mathcal{S}|}{\epsilon} \leq \log_2^2\frac{8H^2|\mathcal{S}|^2}{\epsilon}$$

which let us conclude that

$$m \geq 512\frac{CH^2}{\epsilon^2}(\log_2 \log_2 H)^2\log_2^2\left(\frac{8H^2|\mathcal{S}|^2}{\epsilon}\right)\ln\frac{6}{\delta_1}$$

is a sufficient condition and thus, the statement to show, holds. $\qquad\square$

## C.5  Proof of Theorem 1

*Proof of Theorem 1.* By Lemma 2, we know that the number of episodes where $|X_{\kappa,\iota}| > \kappa$ for some $\kappa, \iota$ is bounded by $6E_{\max}|\mathcal{S} \times \mathcal{A}|m$ with probability at least $1 - \delta/2$. For all other episodes, we have by Lemma 3 that $|\tilde{R}^{\pi_k} - R^{\pi_k}| < \epsilon$. Since, with probability at least $1 - \delta/2$, we have by Lemma 1 $M \in \mathcal{M}_k$, we can use Lemma A.1 which gives $\tilde{R}^{\pi_k} > R^* \geq R^{\pi_k}$ to conclude that with probabilty at least $1 - \delta/2$, for all episodes with $|X_{\kappa,\iota}| \leq \kappa$ for all $\kappa, \iota$, we have $R^* - R^{\pi_k} < \epsilon$. Applying the union bound, we get the desired result, if $m$ satisfies

$$m \geq 512\frac{CH^2}{\epsilon^2}(\log_2 \log_2 H)^2\log_2^2\left(\frac{8H^2|\mathcal{S}|^2}{\epsilon}\right)\ln\frac{6}{\delta_1} \quad \text{and}$$

$$m \geq \frac{6H^2}{\epsilon}\ln\frac{2E_{\max}}{\delta}.$$

From the definitions, we get

$$\ln\frac{6}{\delta_1} = \ln\frac{6CU_{\max}}{\delta} = \ln\frac{6|\mathcal{S} \times \mathcal{A}|C\log_2(|\mathcal{S}|H/w_{\min})}{\delta} = \ln\frac{6|\mathcal{S} \times \mathcal{A}|C\log_2(4|\mathcal{S}|^2H^2/\epsilon)}{\delta}$$

and

$$E_{\max} = \log_2 |\mathcal{S}|\log_2\frac{4H^2|\mathcal{S}|}{\epsilon} \leq \log_2^2\frac{4H^2|\mathcal{S}|}{\epsilon}$$

and

$$\ln\frac{2E_{\max}}{\delta} = \ln\frac{2\log_2 |\mathcal{S}|\log_2(4H^2|\mathcal{S}|/\epsilon)}{\delta} \leq \ln\frac{2\log_2^2(4H^2|\mathcal{S}|/\epsilon)}{\delta}$$

$$\leq \ln\frac{6|\mathcal{S} \times \mathcal{A}|\log_2^2(4|\mathcal{S}|^2H^2/\epsilon)}{\delta}.$$

Setting

$$m = 512(\log_2 \log_2 H)^2\frac{CH^2}{\epsilon^2}\log^2\left(\frac{8H^2|\mathcal{S}|^2}{\epsilon}\right)\ln\frac{6|\mathcal{S} \times \mathcal{A}|C\log_2^2(4|\mathcal{S}|^2H^2/\epsilon)}{\delta}$$

is therefore a valid choice for $m$ to ensure that with probability at least $1 - \delta$, there are at most

$$6mE_{\max} = 3072(\log_2 \log_2 H)^2 \frac{CH^2 |\mathcal{S} \times \mathcal{A}|}{\epsilon^2}$$
$$\times \log_2^2 \left( \frac{4H^2|\mathcal{S}|}{\epsilon} \right) \log^2 \left( \frac{8H^2|\mathcal{S}|^2}{\epsilon} \right) \ln \frac{6|\mathcal{S} \times \mathcal{A}|C \log_2^2(4|\mathcal{S}|^2 H^2/\epsilon)}{\delta}$$

$\epsilon$-suboptimal episodes.

$\square$

## D   Proof of the Lower PAC Bound

*Proof of Theorem 2.* We consider the class of MDPs shown in Figure 1. The MDPs essentially consist of $n$ parallel multi-armed bandits. For each bandit, there exist $m + 1 = |\mathcal{A}|$ possible instantiations, which we denote by $I_i = 0 \ldots m$. The instantiation, or *hypothesis*, $I_i = 0$ corresponds to $\epsilon_i(a) = \mathbb{I}\{a = a_0\}\epsilon'/2$, that is, only action $a_0$ has a small bias. The other hypotheses $I_i = j$ for $j = 1 \ldots m$ correspond to $\epsilon_i(a) = \mathbb{I}\{a = a_0\}\epsilon'/2 + \mathbb{I}\{a = a_j\}\epsilon'$. We use $I = (I_1, \ldots I_n)$ to indicate the instance of the entire MDP.

We define $G_i = \{\omega \in \Omega : \pi(i) = a_{I_i}\}$, the event that $\pi$, the policy generated by $A$ chooses optimally in bandit $i$. For a given instance $I$, the difference between the optimal expected cumulative reward $R_I^*$ and the expected cumulative reward $R_I^\pi$ of policy $\pi$ is at least

$$R_I^* - R_I^\pi \geq (H - 2) \left( 1 - \frac{1}{n} \sum_{i=1}^n \mathbb{I}\{G_i\} \right) \frac{\epsilon'}{2}.$$

For $\pi$ to be $\epsilon$-optimal, we therefore need

$$\epsilon \geq R_I^* - R_I^\pi \geq (H - 2) \left( 1 - \frac{1}{n} \sum_{i=1}^n \mathbb{I}\{G_i\} \right) \frac{\epsilon'}{2},$$

$$\frac{2\epsilon}{(H-2)\epsilon'} \geq \left( 1 - \frac{1}{n} \sum_{i=1}^n \mathbb{I}\{G_i\} \right),$$

$$\frac{1}{n} \sum_{i=1}^n \mathbb{I}\{G_i\} \geq \left( 1 - \frac{2\epsilon}{(H-2)\epsilon'} \right),$$

$$\frac{1}{n} \sum_{i=1}^n \mathbb{I}\{G_i\} \geq \left( 1 - \frac{2\epsilon(H-2)\eta}{(H-2)16\epsilon e^4} \right) = 1 - \frac{\eta}{8e^4}$$

where we chose value $\epsilon' := \frac{16\epsilon e^4}{(H-2)\eta}$ for $\epsilon'$. We will specify the exact value of parameter $\eta$ later. The condition basically states that at least a fraction of $\phi := 1 - \eta/(8e^4)$ bandits need to be solved optimally by $A$ for the resulting policy $\pi$ to be $\epsilon$-accurate. For $A$ to be $(\epsilon, \delta)$-correct, we therefore need

$$\mathbb{P}_I \left( \frac{1}{n} \sum_{i=1}^n \mathbb{I}\{G_i\} \geq \phi \right) \geq \mathbb{P}_I(R_I^* - R_I^\pi \geq \epsilon) \geq 1 - \delta$$

for each instance $I$. Using Markov's inequality, we obtain

$$1 - \delta \leq \mathbb{P}_I \left( \frac{1}{n} \sum_{i=1}^n \mathbb{I}\{G_i\} \geq \phi \right) \leq \frac{1}{n\phi} \sum_{i=1}^n \mathbb{E}_I[\mathbb{I}\{G_i\}] \leq \frac{1}{n\phi} \sum_{i=1}^n \mathbb{P}_I(G_i)$$

All $G_i$ are independent of each other by construction of the MDP. In fact $\sum_{i=1}^n \mathbb{I}\{G_i\}$ is Poisson-binomial distributed as $\mathbb{I}\{G_i\}$ are independent Bernoulli random variables with potentially different mean. Therefore, upper bounds $\delta_i$ must exist such that $\delta_i \geq P_I(G_i^C)$ for all hypotheses $I$ and such that $1 - \delta \leq \frac{1}{n\phi} \sum_{i=1}^n (1 - \delta_i)$ or equivalently $n(1 + \delta\phi - \phi) \geq \sum_{i=1}^n \delta_i$. Since all $G_i$ are independent of each other and

$$\epsilon' = \frac{16\epsilon e^4}{(H-2)\eta} \leq \frac{16(H-2)e^4\eta}{(H-2)64e^4\eta} = \frac{1}{4}$$

we can apply Theorem 1 by Mannor and Tsitsiklis [19] in cases where

$$\delta_i \leq \frac{1}{\eta}(1 - \phi + \delta\phi) \leq \frac{1}{\eta}(1 - \phi + \delta) \leq \frac{1}{8e^4} + \frac{\delta}{\eta} \leq \frac{2}{8e^4}.$$

This result gives us the minimum expected number of times $\mathbb{E}_I[n_i]$ we need to observe state $i$ to ensure that $P_I(G_i^C) \leq \delta_i$

$$\mathbb{E}_I[n_i] \geq \left\lceil \frac{c_1(|\mathcal{A}| - 1)}{\epsilon'^2} \ln\left(\frac{c_2}{\delta_i}\right) \right\rceil \mathbb{I}\{\eta\delta_i \leq 1 - \phi + \phi\delta\},$$

for appropriate constants $c_1$ and $c_2$ (e.g. $c_1 = 400$ and $c_2 = 4$). We can find a valid lower bound for the total number of samples for any $\delta_1, \ldots \delta_n$ by considering the worst bound over all $\delta_1, \ldots \delta_n$. The following optimization problem encodes this idea

$$\min_{\delta_1,\ldots\delta_n} \sum_{i=1}^{n} \ln \frac{1}{\delta_i} \mathbb{I}\{\eta\delta_i \leq 1 - \phi + \phi\delta\} \tag{11}$$

$$\text{s.t. } \sum_{i=1}^{n} \delta_i \leq n(1 + \phi\delta - \phi)$$

As shown in Lemma D.1 in the supplementary material, the optimal solution of the optimization problem in Equation (11) is $\delta_1 = \cdots = \delta_n = c$ if $\eta(1 - \ln c) \leq 1$ with $c = 1 + \delta\phi - \phi$. Since the left-hand side of this condition is decreasing in $c$, we can plug in a lower bound of $c \geq 1 - \phi = \frac{\eta}{8e^4}$ and get the sufficient condition

$$\eta(1 - \ln \frac{\eta}{8e^4}) = \eta(1 - \ln \eta + 4 + \ln 8) \leq 1.$$

It is easy to verify that $\eta = 1/10$ satisfies this condition. Hence $\delta_1 = \cdots = \delta_n = c$ is the optimal solution to the problem in Equation (11). In each episode, we only observe a single state $i$ and therefore, there need to be at least

$$\mathbb{E}_I[n_A] \geq \sum_{i=1}^{n} \mathbb{E}_I[n_i] \geq \frac{c_1(|\mathcal{A}| - 1)n}{\epsilon'^2} \ln\left(\frac{c_2}{\delta_i}\right) \geq \frac{c_1(|\mathcal{A}| - 1)n}{\epsilon'^2} \ln\left(\frac{c_2}{\delta + \frac{\eta}{8e^4}}\right)$$

observed episodes for appropriate constants $c_1$ and $c_2$. Plugging in $\epsilon'$ and $n = |\mathcal{S}| - 3$, we obtain the desired statement.

$\square$

**Lemma D.1.** *The optimization problem*

$$\min_{\delta_1\ldots\delta_n \in [0,1]} \sum_{i=1}^{n} \ln \frac{1}{\delta_i} \mathbb{I}\{\eta\delta_i \leq c\}$$

$$\text{s.t. } \sum_{i=1}^{n} \delta_i \leq nc$$

*with $c \in [0, 1]$ and*

$$\eta(1 - \ln c) \leq 1$$

*has optimal solution $\delta_1 = \cdots = \delta_n = c$.*

*Proof.* Without the indicator part in the objective, we can show that $\delta_1 = \cdots = \delta_n = c$ is an optimal solution by checking the KKT conditions and noting that the problem is convex. Let $k$ denote the number of $\delta_j$ that are set such that the indicator function is 0. Without loss of generality we can assume that their value is $\delta_P := c/\eta$ and the remaining $\delta_j$ take the same value $\delta_A$ (for a fixed $\delta_P$ and $k$, the problem reduces to the one without the indicator functions). Then the problem transforms into

$$\min_{\delta_A \in (0,1), k \in \{0,1,\ldots n\}} (n - k) \ln \frac{1}{\delta_A}$$

$$(n - k)\delta_A + k\delta_P \leq nc$$

We can rewrite the constraint as

$$(n-k)\delta_A + k\delta_P \leq nc$$

$$(n-k)\delta_A \leq nc - k\delta_P = \left(n - \frac{k}{\eta}\right)c$$

$$\delta_A \leq \frac{n - \frac{k}{\eta}}{n-k}c.$$

Since the objective decreases with $\delta_A$, it is optimal to choose $\delta_A$ as large as possible. The optimization problem then reduces to

$$\min_{k \in \{0,\ldots\lfloor n/\gamma \rfloor\}} (n-k)\ln\left(\frac{n-k}{n-\gamma k}c^{-1}\right).$$

where we used for convenience $\gamma := 1/\eta$. We want to show that the optimal solution to this problem is $k = 0$. We can therefore relax the problem to the continuous domain without loss of generality

$$\min_{k \in [0, n/\gamma]} (n-k)\ln\left(\frac{n-k}{n-\gamma k}c^{-1}\right).$$

By reparameterizing the problem with $\alpha = k/n$, we get

$$\min_{\alpha \in [0, 1/\gamma]} n(1-\alpha)\ln\left(\frac{1-\alpha}{c(1-\gamma\alpha)}\right).$$

We realize that the minimizer does not depend on $n$ (while the value does). The second derivative of the objective function is

$$n\frac{(\gamma-1)^2}{(1-\gamma\alpha)^2(1-\alpha)},$$

which is nonnegative for $\alpha \in [0, 1/\gamma]$. Hence, the objective is convex in the feasible region and the minimizer of this problem is $\alpha = 0$ if the derivative of the objective is nonnegative in $0$. The derivative of the objective in $0$ is given by

$$n(\gamma - 1 + \ln(c)).$$

A sufficient condition for $\alpha = 0$ being optimal is therefore

$$\gamma \geq 1 - \ln c$$

or, in terms of the original problem with $\eta = 1/\gamma$, $\delta_1 = \ldots \delta_n = c$ is optimal if

$$\eta(1 - \ln c) \leq 1$$

$\square$