[Reviews · NeurIPS 2015]

Submitted by Assigned_Reviewer_1

***************************************** SUMMARY ***************************************** Paper: Sample Complexity of Episodic Fixed-Horizon Reinforcement Learning

The authors tackle the problem of minimising sample-complexity for episodic fixed-horizon (undiscounted) MDPs. The main contributions are:

(a) a new algorithm, inspired by UCRL\gamma for the discounted setting

(b) a theoretical analysis showing that the new algorithm enjoys sample-complexity guarantees that scale optimally with the horizon, the accuracy and the confidence. The result is sub-optimal only with respect to the size of the state/space, where it scales potentially quadratically while known lower bounds are linear in the state-space.

***************************************** CLARITY *****************************************

Overall I found the paper relatively well executed, but it's clear the authors struggled to include as much as possible in the main paper, and the readability suffers as a consequence. I don't know how to feel about this. On the one hand it's better to include proofs in the main body (this is a theory-only paper). On the other hand, as a standalone paper it is still relatively hopeless to verify the correctness of the

results. Anyways, I read enough of the supplementary material to be convinced that the results are correct.

***************************************** QUALITY, ORIGINALITY AND SIGNIFICANCE *****************************************

Things I liked:

* The setting is quite a bit cleaner than the discounted setting where there are no clear demarcation points to reset the policy

* The results are relatively tight. Improving the dependence on the state-space would be nice, of course, but there are many cases where H^3 is larger than |S| and so far there are no algorithms with linear dependence on |S| in this setting (presumably it is not hard to adapt MORMAX, however).

* The trick to allow C > 2 in the algorithm. (for me this is the main novelty) By the way, maybe it is worth to look at [2] and see if the Good-Turing trick can (has been?) be used to solve this problem.

Things I disliked:

* The proof is following more-or-less exactly as the same cumbersome proof of Lattimore & Hutter, even using the recursive substitution of the higher-moment value estimates. I wish there was a simple proof of this result and I think more time could have been taken to look for one.

* The novel contribution is on the low side. The setting is different, but intuitively there is not so much difference between the discounted and episodic settings with the latter usually being a little easier. Of course it is still a lot of work to check this, but many results go through with little modification.

* The paper is a bit of a struggle to read.

* The algorithm (like UCRL\gamma) is less principled than it could be. Eg., samples are not used as soon as they are obtained, and samples obtained beyond a certain point are simply discarded. I understand these tricks are useful to make the proof work, but my suspicion is that they are going to make the algorithm fairly impractical. I think many of the union bounds can be replaced with uniform bounds as used in [3] (and elsewhere). Perhaps this could be investigated in the future.

Overall I am inclined to accept. I hope the authors will try (hard) to improve the readability to a slightly less specialised audience.

Minor Comments:

Line 51: Perhaps mention here (as in the abstract) that the upper and lower bounds do not match on the size of the state-space Line 99: You want to say that if the set of rewards are *not* known, then you can do this transformation Line 151: "as phase" -> "as a phase"? Often: When I see "infinite horizon" I am usually thinking of undiscounted average reward setting, but here you are mostly using it the discounted setting.

Line 154: URCL-\gamma -> UCRL-\gamma? Line 223: Did Kakade really exploit the low variance for MDPs? Perhaps you mean the work by Azar [1], which should probably be cited (it is using a generative model and discounting, so different, but still very related) Line 270: "current policy" -> "the current policy" Line 304: This proof sketch is basically impossible to follow without just reading the proof in the appendix. Perhaps better to give a more intuitive sketch like the previous lemma and just refer the reader to the appendix?

[1] On the Sample Complexity of Reinforcement Learning with a Generative Mode. Azar, Munos & Kappen. 2012 [2] PAC Optimal Planning for Invasive Species Management: Improved Exploration for Reinforcement Learning from Simulator-Defined MDPs. Dietterich, Taleghan & Crowley [3] lil'UCB : An Optimal Exploration Algorithm for Multi-Armed Bandits. Jamieson, Malloy Nowak & Bubeck

Summary: The paper is reasonably well written, but still quite hard to get through. The results are good. There is a modest novel contribution.

Submitted by Assigned_Reviewer_2

---Paper Summary---

This paper presents a novel model-based RL algorithm for the fixed-horizon case with PAC bounds on the sample complexity (tighter than previous results). Additionally a novel lower bound on the sample complexity in this setting is derived.

---Quality---

The primary contributions of this paper are two theorems (upper and lower bounds). I must admit that I did not read the prodigious appendix in full detail, so I cannot confidently assert the correctness of those proofs. However, I do find the sketch argument provided in the main paper to be sensible and plausible.

---Clarity---

The authors have done an admirable job of clearly setting up a highly technical contribution, and also very clearly drawing connections to and distinctions from existing work. The main clarity concern is that the full proof of the main results is relegated to the 14 page appendix. Personally, I have concerns about the community-level implications of accepting papers for which the substantial majority of the technical content is "supplementary" and, in principle, not peer-reviewed. At some point, if your paper is 22 pages long, maybe an 8 page conference paper is not the best vehicle to communicate your ideas. However, in the absence of clear guidelines, I defer to higher levels of decision-makers to decide at what point the balance of content between the main paper and appendix becomes inappropriate.

In the case of this paper, I genuinely appreciate the author's efforts to outline the argument at a high level, and to point out the main distinctions between it and other similar arguments. Considering the space constraints, I think they've made a valiant effort to clearly communicate their main ideas. Despite the authors' efforts, I do fear that this discussion will be very hard to follow for any reader not already immersed in the relevant PAC-RL literature, since it relies pretty heavily on the reader being able to swap in various components to the standard argument template.

Finally, this is a minor and purely stylistic point but I would like to discourage the authors from using so many footnotes. Footnotes break the flow of a paragraph (and the concentration of the reader); they should be used sparingly. They are also rarely necessary. Most of the footnotes in this paper would work better as simply another sentence in the paragraph. Also, sometimes the footnote indicator appears in a poor location (e.g. footnote 5 comments on a statement that appears after the footnote indicator).

---Originality---

To my knowledge the algorithm and results presented here are novel. The algorithm and analysis do bear many similarities to existing PAC-RL work, but this is typical for this area. The analysis is not a trivial application of another result, and did require additional technical insight. I thought the authors did a very good job of comparing and contrasting their work to the relevant work in the literature.

---Significance---

I think the authors make a good case for the practical importance of the fixed horizon setting. As is often the case with PAC-RL results, it is not clear that the presented algorithm will have practical impact. Also, most of the key insights driving the analysis have already been obtained in other results so it's not clear that these results have much impact on our conceptual/theoretical understanding of the exploration/exploitation problem. That said, the careful application of these ideas to the finite horizon case is non-trivial, and obtaining tighter bounds does help to reveal they key issues at play in that particular case.
Summary: I think the work is sound, the problem is of sufficient interest, and the paper is mostly well written. My main complaint is just that this is a 25-30 page paper in an 8 page paper's body and that does impact the clarity of the presentation and the potential breadth of the audience. The authors do work hard to get their key ideas across as clearly as they can within the main paper, so I lean toward accepting this paper, albeit with some reluctance.

Submitted by Assigned_Reviewer_3

The authors provide upper and lower PAC bounds for episodic, fixed-horizon RL. The new bounds are much tighter than related work in terms of horizon length. The upper bound is presented with an algorithm, and the lower bound is presented with a class of MDPs that attains the bound. The technical tools are extensions of work by Lattimore and Hutter uses a finer grained notion of when a transition is "well-estimated."

I have two questions:

What is the relationship between the fixed-horizon EVI and work by Mannor et al. about finding minimax policies within a confidence set?

At the end of the day, what is the time complexity of Algorithm 1?

Quality: To the best of my knowledge, the results are sound Clarity: The paper is well-written. I have minor comments below. Originality: The paper represents a novel, non-trivial use of cutting-edge tools for analyzing sample complexity of RL algorithms. Significance: Finite-horizon problems have wide applicability as data analysis tools; the authors mention automated tutoring and there are also healthcare applications as well. The paper is significant because it gives a better understanding of the data demands of solving such problems, and because it presents an algorithm that could be used or extended for applied problems in this area.

Minor comments: -059: "has generally focused on slightly alternate situations" -- I think either "employed different strategies" or "considered different settings". Review and re-word for clarity. -087: I think p0 is unused. -140:

"that enables to obtain" -- "that enables us to obtain" or re-word. -145: "e.g." -- "e.g.," -248: "knowness" -- "knownness" (check throughout) -252: "expected frequence" -- "expected frequency" or count or similar -263: What is the $m$ in the denominator? -366: "The range of possible..." I think this is a run-on sentence; re-word for clarity.
Summary: The authors provide upper and lower PAC bounds for episodic, fixed-horizon RL. The new bounds are much tighter than related work in terms of horizon length.

Submitted by Assigned_Reviewer_4

This paper extends some of the recent PAC exploration bounds from the discounted, infinite horizon setting to the episodic fixed-horizon setting.

---------------------------- Quality: ----------------------------

Overall this paper is well written, concepts are well explained and claims are rigorously proven. My main concern is with lemma 3: How can it be that we only care about the number of state-action pairs with low knowness? Consider the case where very few state-action pairs have low knowness (just one in the limit), but they have very high importance (almost all the weight is concentrated on them). I don't see how one can guarantee that an episode based on such a policy would be expected to have close to optimal accumulated reward. Am I misunderstanding the definition of knowness? Is there an unstated assumption that this situation cannot happen for UCFH? Is the lemma incorrect?

I have some more questions/comments of lesser relative importance in my line by line comments.

---------------------------- Significance: ----------------------------

I view this as a purely theoretical contribution. Proving better theoretical bounds is a positive contribution, but I doubt this is an algorithm that people would want to implement and use in a practical application. My hope is that this paper will spark future work leading to a practical algorithm with the same or better bounds.

---------------------------- Originality: ----------------------------

Both the algorithm and analysis presented in this paper borrow heavily from the work of Lattimore and Hutter. Having said that, I believe that extending those results to the episodic, fixed-horizon setting was far from trivial.

---------------------------- Clarity: ----------------------------

The quality of exposition in the main body of the paper is above average for this type of paper. The authors offer some very useful intuition behind their proof. On the other hand, the appendix is quite dense (though once again, this is not atypical for this type of paper).

Line by line comments:

Line 55: to wide -> to a wide Line 120: of s and action a -> of state s and action a Line 160: Why is the transition probability time dependent? The chosen action (policy) will be time dependent, but unless I'm missing something the transition probabilities themselves will not be time dependent. Algorithm 1: Please include n(s,a) and S(s) in the initialization. Theorem 1: Please spell out the exact complexity in the theorem. You can choose to give both the exact and \tilde O forms, but you cannot expect the reader to have to go through your proofs to reconstruct the exact form. Many of the terms dropped by \tilde O can be much larger than H in domains of realistic size. Line 269: The way the active set is defined, isn't it the case that state-action pairs not in the active set are "completely" and not just "very" unlikely under the current policy? Can a state-action pair with non zero weight have zero importance? Line 315: . where -> , where Line 371: at least -> at least a Section 5: Please use the same ordering for |S|, |A|, and H in all bounds.

Line 426: to wide -> to a wide Line 541: Why exactly one? Couldn't 1 to H (s,a) pairs cross the threshold in the same episode? (If this is the case, you will also need to change your wording in Algorithm 1) Line 592: for a details -> for details Line 644: Make "proof of lemma 2" bold Line 722: fix -> fixed

Post discussion comments: -------------------------------------------------------------------------- The authors have adequately addressed my concerns in their response, and I have adjusted my score accordingly.
Summary: This is a fairly well written paper, extending recent theoretical results to a reinforcement learning setting that has received far less attention.

Author Feedback
Author rebuttal: We thank all reviewers for thoughtful comments, which are in general positive about clarity, quality and significance of our work. We will revise our paper accordingly and, we here focus on addressing the main concerns.

R1, R7: Clarity / Balance between main paper and appendix
We appreciate the reviewers' preference that as much as possible the technical details are in the main paper. We will revise our paper to do our absolute best to make the paper both as clear and as self contained as possible, but like many past NIPS papers with detailed theoretical content, we think it is important to include all the proof details in a comprehensive form which requires the appendix.

R2: Time complexity of our approach
After at most \tilde O(H^3 |S|^2 |A|^2 C / eps^2 ln(1 / delta)) computation steps, the policy of UCFH is eps-optimal with sufficiently high probability. We will include the runtime- and space-complexity analysis of UCFH.

R2: Relation of FH-EVI to previous work:
Indeed our EVI planning approach can be viewed as the optimistic counterpart to robust MDP planning (e.g. Xu and Mannor (2010) and Delage and Mannor (2007)). We are happy to add a sentence about this connection.

R3: Correctness of Lemma 3:
R3 asked if a counterexample to Lemma 3 would be if there are very few high-importance state-action pairs that have low knownness as the conditions only are concerned with the number of low-knownness sa-pairs but not (directly) their importance.
The key insight to why such a situation is not an issue is that knownness and importance are coupled as both definitions include the weight w(s,a).
Assume the extreme case that only a single (s,a)-pair has very high importance. Its weight w(s,a) is then close to the horizon H. There are two cases of its knownness kappa:
1. Kappa = 0. This is impossible, due to the condition in Lemma 3: e.g. such a (s,a) pair cannot exist. Therefore kappa must be > 0.
2. Kappa > 0. Due to the definition of knownness, (s,a) must be observed at least n(s,a) >= mH times in this case. The lower bound on m stated in Lemma 3 is large enough to ensure that the transition probabilities are sufficiently well estimated. This in turn implies that the values in the different MDPs are close enough to allow for close to optimal accumulated reward, which addresses the stated concerns.

Now consider that the total weight is distributed among several s-a pairs. For a good value estimate, we do not need to estimate all transitions equally well. The smaller their weight, the more sa-pairs can have less accurate estimates of their trans. prob. The |X_ki| <= k condition in Lem. 3 (similar to Lattimore & Hutter 2012) encodes this insight as the definition of knownness includes the weight.

R3: Time-dependent transition probability (re. Line 160):
The reviewer is completely correct that the true transition model is assumed to be stationary. We defined M_k as allowing time-dependent transition probabilities just for the purposes of optimistic planning (since the most optimistic transition parameters in the possible set will likely change based on the time step in finite horizon settings). The theoretical results hold whether we define M_k as having a stationary or time-dependent transition model as the true MDP M is with high probability in M_k in both cases.

R3: SA-pairs not in the active set are impossible (re. Line 269):
Thanks for catching this-- definition of the active set contained a typo. Instead of iota(s,a) > 0, the condition should be w(s,a) >= w_min..

R3: Several sa-pairs crossing the threshold (re. Line 541):
To simplify the theoretical analysis, we assume that when multiple sa-pairs cross the threshold simultaneously, a policy updates happens for each of them individually. In actual implementations, the policy would just be computed once incorporating the new statistics of all sa-pairs that crossed the threshold without harming the PAC bound. We will clarify this.

R4: Derive improvements for infinite H / L&H algorithm?
Yes, our work could help improve such through extending them to handle non-sparse transitions (C > 2) which reduces the gap between algorithms with good theoretical guarantees and ones that are practical.

R4: Main difference episodes, not finite horizon
Indeed, our work could be extended easily to other episodic settings (e.g. discounted rewards). For the sake of clarity we focused on a specific setting but we will add a remark about other episodic formulations.

R7:
We thank the reviewer for the many useful suggestions and will do our absolute best to improve readability of the paper using the feedback from all reviews. Though our work does build on prior work, extending it to the finite horizon case required a number of non-trivial insights and, as R1 also noted, we think tackling the episodic finite horizon case which is important because it is more suitable to many real world applications than discounted rewards.